**communications** engineering

# Fabrication of cell culture hydrogels by robotic liquid handling automation for high-throughput drug testing
Eloisa Torchia[1], Moises Di Sante [1], Bohdana Horda[1], Marko Mihajlovic[2], Julius Zimmermann[1], Melissa Pezzotti[1], Elisa Cimetta[3], Sylvain Gabriele [4], Ferdinando Auricchio[5], Johan Ulrik Lind[2], Alessandro Enrico [1] ✉ & Francesco Silvio Pasqualini [1] ✉

Traditional plastic and glass culture lacks physiological relevance, undermining predictive power in drug discovery. Organoids and organs-on-chip improve biomimicry but do not scale to high-throughput screening (HTS). Even simple hydrogel coatings in HTS plates suffer from curved menisci that disrupt seeding and imaging. We present HYDRA (HYDrogels by Robotic liquid-handling Automation), an automated method to fabricate thin, planar hydrogel films directly in standard plates. Liquid-handlers dispense sub-contact volumes without wall wetting; immediate re-aspiration pins the contact line, leaving a uniform layer with controlled stiffness and thickness. Using fish gelatin hydrogel, HYDRA produces meniscus-free coatings compatible with routine 96- and 384-well workflows and plate-scale quality control. HYDRA was validated through imaging-based dose-response assays with anticancer compounds, engineered epithelial monolayers, and long-term holographic and fluorescence microscopy. It preserved pharmacological sensitivity while supporting high-content imaging on soft, biomimetic substrates, offering a practical bridge between physiological relevance and HTS scalability for early in-vitro drug testing.

Accurately predicting human cellular responses at preclinical stages is essential for successful drug R&D. Yet, drug discovery programs often fail (90% of the time), progress slowly (10+ years), and are costly (1 BUSD + )[1]. Advances in genomics and artificial intelligence have accelerated early-stage discoveries[2,3], but productivity continues to decline, and clinical attrition rates remain high[4,5]: ~50% of compounds that pass initial preclinical assays still fail in subsequent human trials, reflecting the poor predictive capacity of traditional preclinical models[1,6]. This is perhaps not surprising since cell-based assays, especially early automated high-throughput screening (HTS), have predominantly relied on rigid substrates such as plastic and glass, which distort tissue architecture, alter mechanotransduction and cellular metabolism, and bias cell-ECM interactions, thereby compromising physiological assay relevance relative to compliant 2D hydrogel beds[7–10].

The effect of improved biomimicry on predictivity has been demonstrated by the success of engineered culture systems that replicate the native tissue microenvironment more closely than standard cell culture formats[11].

However, these systems haven't reached the throughput necessary to be effectively employed in early HTS. Spheroids and organoids both provide three-dimensional cell–cell and cell–ECM interactions, accurately recapitulating tissue architecture and clinically relevant disease phenotypes[12]. Their potential compatibility with multiwell plates also makes them attractive candidates for HTS[13,14]. However, their size, stochastic structural organization, and inherent optical scattering strongly limit quantitative high-resolution microscopy, complicating routine use in imaging-based HTS[11]. Moreover, organoids typically lack integrated vasculature, necessitating additional complex microfluidic engineering[15–17]. Organs-on-chips similarly capture complex tissue functionality through precise control of mechanical and biochemical conditions within microfluidic platforms[12]. Their planar configuration facilitates high-resolution microscopy and optical clarity. Yet, the intricate and custom nature of microfabrication currently prevents scalable production, severely restricting their practical implementation within automated, high-throughput pipelines[11]. In parallel

---

[1]E. Torchia, M. Di Sante, B. Horda, J. Zimmermann, M. Pezzotti, A. Enrico, F. S. Pasqualini Synthetic Physiology Lab, Department of Civil Engineering and Architecture, University of Pavia, Pavia, Italy. [2]M. Mihajlovic, J. U. Lind Department of Health Technology, Technical University of Denmark, Lyngby, Denmark. [3]E. Cimetta Department of Industrial Engineering, University of Padua, Padova, Italy. [4]S. Gabriele Mechanobiology & Biomaterials Group, Research Institute for Biosciences, University of Mons, CIRMAP, 20 Place du Parc, Mons, Belgium. [5]F. Auricchio Group of Computational Mechanics and Advanced Materials, Department of Civil Engineering and Architecture, University of Pavia, Pavia, Italy. ✉e-mail: alessandro.enrico@unipv.it; francesco.pasqualini@unipv.it

with 3D organoids and microfluidic organs-on-chips, microfabricated 2.5D platforms have matured to engineer laminar, quasi-three-dimensional tissues on planar substrates that preserve optical access and enable integrated functional readouts. Notable examples include multimaterial-printed cardiac devices with embedded strain sensors for continuous mechanical measurements[18], high-throughput electrode arrays enabling electrical phenotyping across many conditions[19,20], and microarray platforms with tunable substrate stiffness for parallel mechanobiology assays[21]. Collectively, these 2.5D strategies improve control of geometry, mechanics, and electrophysiology, but often rely on bespoke microfabrication workflows or specialized materials that complicate scale-up in standard HTS pipelines.

Planar hydrogel substrates could provide an advantageous balance between physiological relevance, imaging clarity, and HTS scalability. Hydrogels are widely utilized in organoids and chips, as they offer adjustable biochemical composition, substrate stiffness, and optical transparency[22–24], essential for accurate mechanobiology studies[25]. Achieving uniform thin hydrogel layers within multiwell HTS plates remains technically challenging: the meniscus at well edges creates curvature that disrupts cell attachment, homogeneous growth, and high-resolution microscopy[26,27]. PEG- and hyaluronic acid–based hydrogels have been fabricated in 96-well plates through photocrosslinking, yielding tunable stiffness and viscoelasticity, yet the thick (500–1000 μm) gels compromise optical resolution[28,29]. Machillot et al. established an automated LbL process in microplates, yielding optically transparent PEM films with thicknesses from a few tens of nanometers to a few micrometers, aided by plate tilting and ~10% over-aspiration to improve uniformity[30]. However, effective mechanical shielding of adherent cells from the rigid polystyrene generally requires soft layers above mechanosensing length-scales, on the order of ~10–20 μm for single cells and considerably thicker for colonies (half-maximal depth response of ~54 μm at 1 kPa)[31]. Scaling LbL to such thicknesses in 96/384-well formats would entail impractically many deposition/rinse cycles with elevated risk of defects and delamination; moreover, increased chemical crosslinking (e.g., EDC/NHS) stiffens PEMs and counteracts the desired compliant interface. However, as with Brooks et al., the resulting gels are several hundred micrometers thick, which restricts high-resolution imaging to planes near the well bottom and makes them unsuitable for 2D mechanobiology assays that require a flat, optically accessible surface[28]. Commercial 2.5D and 3D systems highlight the same trade-off: pre-cast PEG matrices (e.g., TrueGel 3D-HTS, 0.5–1 mm) and bioprinted constructs such as RASTRUM (Inventia Life Sciences) enable high-throughput encapsulation of cell-laden gels but remain hundreds of micrometers thick and optimized for volumetric or biomarker assays rather than imaging cells on the gel surface[28,32,33]. Together, these approaches demonstrate the lack of a platform that combines optical flatness and biomimicry in a high-throughput format, while maintaining compatibility with standard plate geometry and imaging (Supplementary Table S1).

Here, we introduce HYDRA (HYDrogels by Robotic liquid-handling Automation), a scalable and automated solution to generate uniform, micrometric planar hydrogels directly within standard HTS plates. HYDRA works by robotically dispensing a sub-contact volume of hydrogel precursor solution, carefully preventing it from contacting the well sidewalls. Immediate controlled re-aspiration pins the solution's contact line through contact angle hysteresis, yielding a thin, uniform precursor layer.

Among available hydrogel chemistries, we selected cold-water fish gelatin (FG) for its favorable handling and biological compatibility. FG remains fluid at room temperature, allowing precise robotic dispensing and re-aspiration, while microbial transglutaminase (TG) enables mild enzymatic crosslinking without UV or heating[34,35]. Unlike synthetic polymers such as PEG that require surface functionalization, gelatin is inherently cell-adhesive, supporting rapid attachment and spreading[36,37]. Gelatin is also an order of magnitude less expensive than common GelMA or collagen (~€0.5 g-1 vs €100s g-1 for GelMA/collagen, current catalog prices from suppliers), an important factor for large-scale screening. These rheological, biochemical, and economic advantages make FG well-suited to form planar, micrometric hydrogel films reproducibly within standard multiwell plates.

Enzymatic crosslinking of this residual layer produces flat hydrogel films of 10–50 μm thickness (Fig. 1a, b). We validated this approach using engineered fluorescent HaCaT epithelial cells cultured in both 96- and 384-well plates (Fig. 1c–e), demonstrating robust adherence, normal proliferation, and suitability for high-resolution, imaging-based drug-response assays.

Thus, HYDRA leverages established HTS automation and microscopy infrastructure, delivering physiologically relevant, easily implementable substrates to pharmaceutical screening programs without new capital investments. Importantly, HYDRA targets image-based drug screening assays, where the combination of biomimetic substrates and high-resolution microscopy can greatly improve predictive power. While its current configuration is not designed for bulk RNA/protein extraction, it addresses the pressing need for imaging-compatible, high-throughput culture formats.

## Results
### Rheological studies of FG hydrogels
To implement HYDRA within automated HTS pipelines, we first needed to verify that FG hydrogels meet essential mechanical and dispensing requirements. We crosslinked FG chemically using TG to fabricate hydrogel-based cell culture platforms with HYDRA (Fig. 2a)[38,39]. Critically, we selected FG because aqueous solutions of FG remain liquid at room temperature (RT) and must be cooled to 4 °C to form a physical gel. The viscosity of FG should thus stay low and stable at RT to make dispensing possible[40]. We first used oscillatory rheometry to optimize hydrogel casting in 96-well plates and to verify FG's compatibility with robotic dispensing by quantifying its viscosity at RT. The viscosity at zero shear rate measures the material's viscosity at rest, and we observed this measure doubling in our precursor solution when we doubled (from 0.67 ± 0.18 to 2.11 ± 0.06 Pa·s) the FG concentration, in good agreement with the literature (Fig. 2b)[39,41]. Still, unlike gelatin solutions obtained from mammalian sources, it remained below the threshold for pipettability with air displacement pipettes[42–44].

The gelation point of the FG hydrogels is expected to decrease with higher polymer concentration[39]. To test FG pipettability across a more extensive range of conditions, we examined FG physical gelation using the tube inversion method. Specifically, we prepared three FG concentrations (5% w/v, 10% w/v, and 20% w/v in PBS -/-) to generate gels with physiological stiffness when mixed with TG[41]. We dispensed FG solutions into microcentrifuge tubes, placed them at RT and 4 °C for 72 hours, and took pictures after 24 hours upon inversion (Supplementary Fig. S1). All the mixtures flowed with gravity except for the 20% FG mixture at 4 °C, for which viscosity was high enough to counter the gravitation pull. Furthermore, we analyzed chemical gelation by crosslinking gelatin solutions with 2% w/v TG at RT and 4 °C for 72 h. Independent of temperature and concentration, all FG solutions formed gels within 72 h (Supplementary Fig. S1). These results show that, in principle, a liquid-handling robot can dispense FG solutions at room temperature.

To ensure gelation occurred only after completing automated dispensing across the entire plate, we evaluated gelation onset using previously reported TG concentrations optimized for bioprinting (0.5% and 2% w/v)[45]. Both TG concentrations resulted in a gradual crosslinking process (Fig. 2c-d). Specifically, the storage modulus (G') and loss modulus (G") curves associated with 0.5% w/v TG suggested that crosslinking proceeded more slowly at this concentration (Fig. 2c). In comparison, the curves for 2% w/v TG reached a plateau at approximately 3 h after mixing (Fig. 2d). For 10% w/v FG hydrogels crosslinked with 2% w/v TG, the crossover point was reached after approximately 40 min, suggesting that rapid blending of the two components in an HTS plate format should not trigger partial gel formation.

Next, we evaluated whether the selected concentrations yielded substrates with stiffness within the physiological range by measuring the shear modulus of the hydrogels (Fig. 2e–f). Samples crosslinked with 2% w/v TG exhibited G' ranging from 500 Pa to 3 kPa (Fig. 2e and Supplementary Fig. S2), again in good agreement with the literature[39].

**Fig. 1 | HYdrogel Dispensing method with Robotic Automation (HYDRA) for HTS-compatible hydrogels. a** Schematic illustration of the Hydrogel Dispensing with Robotic Automation (HYDRA) method. **b** Photograph of the robot performing the HYDRA of fish gelatin (FG) hydrogels in a 96-well plate. A red food colorant was added for visualization purposes. **c** Isometric confocal z-stack (3D rendering) of an enzymatically crosslinked FG hydrogel embedding fluorescent beads and HaCaT cells seeded on top and on the side of the hydrogel. **d** Schematic representation of drug tests performed with nocodazole and paclitaxel. Drug dose (three concentrations of each drug color-coded as blue – low concentration, red – medium concentration, and black – high concentration) increases along columns. **e** Gel compatibility with standard imaging approaches in high throughput plates using HaCaT cells. (i) Time-lapse holographic imaging for long-term low phototoxicity studies and (ii) static fluorescence imaging of cells treated with nocodazole or paclitaxel drugs and imaged for 48 hours after treatment. In both cases the gels were cast on inexpensive traditional tissue culture plastic. iii) High-resolution fluorescence imaging experiments of HaCaT cell proliferation for 18 hours (high magnification, 40× silicon oil objective). The white, green, cyan, and magenta colors in (ii) and (iii) correspond to actin, tubulin, and nuclei of cells in the G1 and S/G2 phases, respectively. Scale bars: 25 μm.

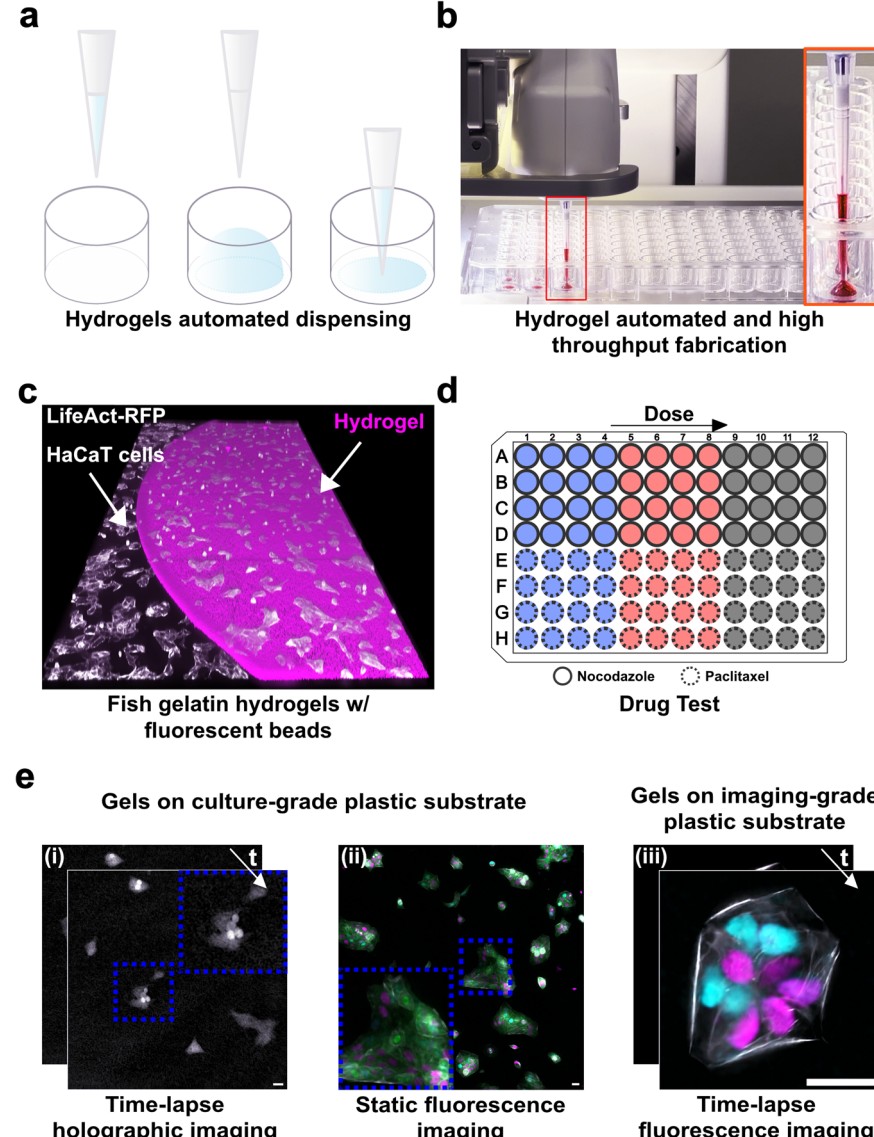

Finally, to understand the effect of TG on the resulting hydrogel stiffness, we used a gelatin concentration of 10% w/v and varied TG concentrations from 0.5 to 2% w/v (Fig. 2f, Supplementary Fig. S2c-d, and S3). We found that the value of G' increased (from 500 Pa to approximately 2 kPa) with increasing TG concentration (Fig. 2f and Supplementary Fig. S3). This trend and the G' values align with previous works reporting on FG hydrogels chemically crosslinked at 37 °C[46]. Together, these results demonstrated that the viscosity of FG at room temperature can be managed using a liquid-handling system, even with high protein concentrations (20% w/v). We also found that the dispensed hydrogels reached stiffness ranges (E~1.5–6 kPa) within the physiological range from single-digit to tens of kPa reported for skin tissues[47] and are consistent with conditions of stiffness already used for HaCaT keratinocytes[48,49].

## Modeling, fabrication, and characterization of hydrogel thin film in 96-well plates

To optimize hydrogel fabrication by minimizing meniscus formation and maximizing substrate coverage, we used finite-element modeling (FEM) in COMSOL. First, we simulated a classical meniscus formation scenario in which the dispensed volume caused wetting of the sidewalls (Fig. 3a; Supplementary Movie S1). Then, we computed the ratio between the radius of the hydrogels with the air-liquid interface parallel ( ± 3°) to the substrate and the radius of the well. This configuration led to the gel covering the entire area of the well (100% radial coverage), but the high-curvature meniscus severely limited the available planar area (10% flatness). Next, we reduced the dispensed volume to avoid wall contact and applied a receding-angle correction to simulate contact-line pinning during re-aspiration (Supplementary Movie S2)[50]. To identify the optimal dispensing volume, we tested four volumes (6, 12, 18, and 24 μL). Without wall contact, both gel coverage and flatness improved with larger volumes, peaking at 18 μL. At 24 μL, however, wall wetting reduced flatness substantially. To account for fabrication non-idealities, we also introduced small fluctuations in the simulation parameters: ±20% on dispensed volume; −100 μm and −200 μm on the well radius because of the fabrication tolerance of the well plate and the accuracy of the robot. Considering these fluctuations, we verified that dispensing 18 μL could result in wall wetting (Supplementary Fig. S4). Therefore, we selected 12 μL for the experiments in this study, which reduced coverage by ~20% and increased flatness by 4.5 times concerning the meniscus-inducing conditions (Fig. 3 b and Supplementary Fig. S4).

Accordingly, we automated FG hydrogel fabrication in 96-well plates using an Integra liquid-handling robot, dispensing and immediately re-aspirating 12 μL of precursor solution (visualized using red colorant) to leave only a thin boundary layer. This procedure required about 10 minutes, well within the 40-minute gelation threshold established earlier (Fig. 3c;

**Fig. 2 | Rheology of fish gelatin hydrogels.**
**a** Schematic of FG hydrogels chemically crosslinked with TG. Glutamine residues in proteins can form covalent bonds with lysine residues through a transamidation reaction, which produces ammonia as a byproduct, which is washed away during the rinsing steps. In this process, the γ-carboxamide groups of glutamines serve as acyl donors, while the ε-amino groups of lysine function as acyl acceptors. Created with Biorender.com. **b** Viscosity vs. shear rate for three different concentrations of FG alone (5% - orange squares, 10% - purple dots, and 20% w/v - black diamonds). Dashed lines stand for viscosity values of 85% glycerol and deionized water at room temperature. **c** G' and G'' vs. time for FG at 10% w/v concentration crosslinked with 0.5% w/v TG. **d** G' and G'' vs. time for FG at 10% w/v concentration, crosslinked with 2% w/v of TG. Data are displayed as mean ± s.e.m ($n$ = 3 samples). **e** Storage modulus G' vs. FG concentration (5% - orange squares, 10% - purple dots, and 20% w/v - black diamonds). FG solutions were crosslinked with 2% w/v TG. Data are displayed as mean ± s.e.m ($n$ = 3 samples). **f** Storage modulus G' vs. TG concentration (0.5% - orange squares, 1% - purple dots, and 2% w/v - black diamonds). 10% w/v fixed FG was used. All data are displayed as mean ± s.e.m ($n$ = 3 samples).

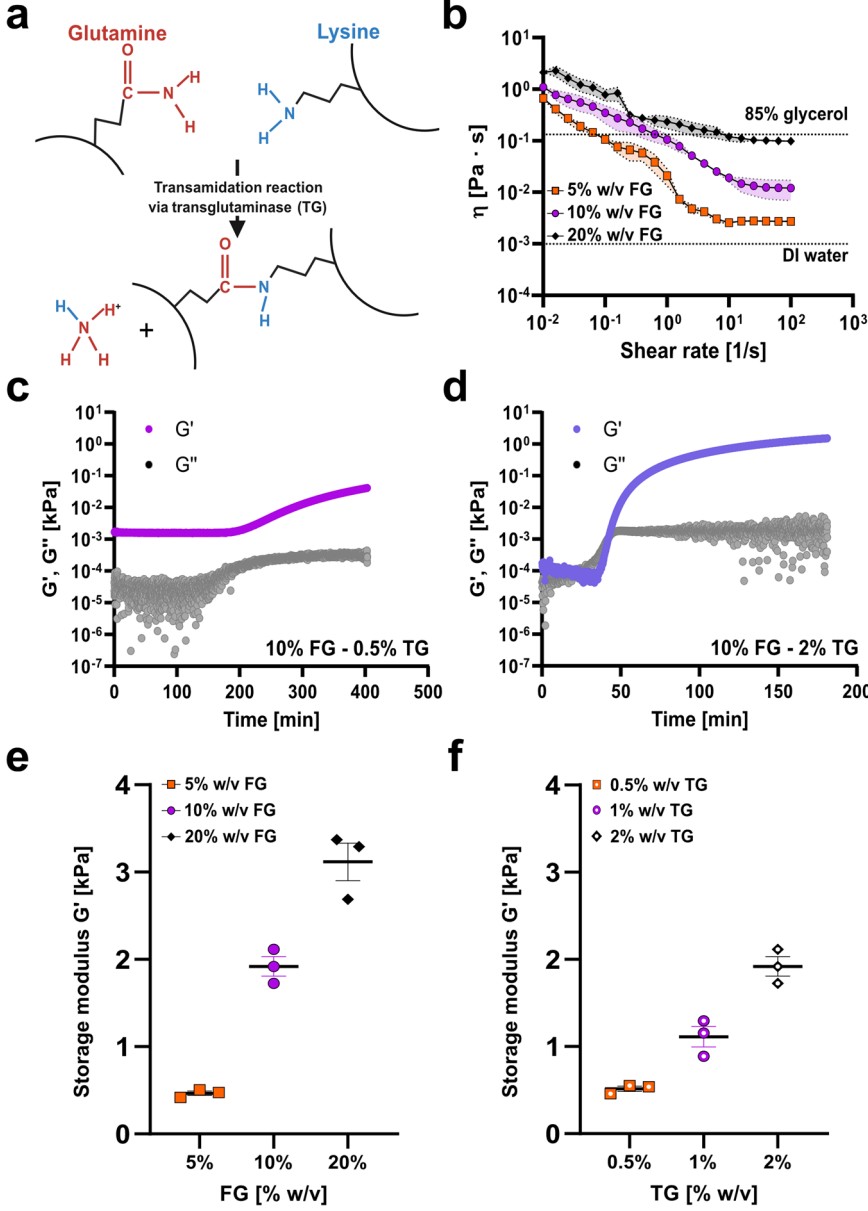

Supplementary Movies S3 and S4). To experimentally visualize the meniscus of the liquid, we imaged full meniscus geometries in z-stacks across high-throughput wells (Supplementary Fig. S4d–e). These reconstructions, shown in orthogonal planes (xy, xz, yz) and at higher magnification in the xy view, highlight the extent of curvature and its impact on planar coverage, with cellular actin in gray and fluorescent bead–embedded hydrogels in magenta.

Next, to verify compatibility with standard microscopy, we characterized hydrogel thickness and flatness by embedding fluorescent beads within the gels and seeding epithelial cells expressing a red fluorescent actin marker (LifeAct™) (Fig. 3d). Using a 20× air objective (NA 0.75), we captured two images, one at the top of the gel and the other at the bottom of the well, by focusing on cell actin and the fluorescent beads. The samples exhibited an expected dose-dependent increase in thickness, from ~10 μm to ~30 μm, and intergroup reproducibility (Fig. 3e and Supplementary Fig. S5). To assess whether this effect was linked to gel swelling, we also measured dry gel thickness using optical profilometry (Supplementary Fig. S5c–d). The resulting swelling ratios decreased with increasing FG concentration ( ~ 6.7 at 5%, ~4.1 at 10%, and ~1.1 at 20% FG), indicating that the greater hydrated thickness of 20% gels mainly reflects their higher

solid content rather than swelling. This result agrees with the literature, which sets the cutoff point for how deeply cells sense substrates at 10-20 μm[31,51,52]. Together, these results support the idea that liquid-handling automation can reproducibly form thin hydrogel layers for cell culture in commercial HTS plates.

## Drug testing using digital holography for long-term imaging
To demonstrate HYDRA's utility in drug testing, we sparsely seeded HaCaT epithelial cells on gels and treated them with nocodazole and paclitaxel, two well-characterized cell cycle inhibitors[53–55]. The primary aim was to highlight the feasibility of integrating hydrogel-based cell culture platforms into imaging-based drug screening systems. As a stress test, we used long-term digital holography—an imaging method with minimal phototoxicity but sensitive to subtle refractive index shifts[56] to determine whether HYDRA's hydrogel layers interfered with detection. Having added a thin hydrogel layer to provide a more biomimetic microenvironment for the cells with HYDRA, we asked whether digital holography could still detect a sufficiently large refractive index change. Furthermore, we used cell-culture plastic 96-well plates for this stress test, which have poorer optical properties than the HTS plates designed for

**Fig. 3 | Hydrogel Dispensing with Robotic Automation (HYDRA) fabrication method.** 2D representation of hydrogel profile angles resulting from COMSOL simulation after (**a**) conventional casting, displaying meniscus formation, and (**b**) HYDRA casting. (**c**) Sequential frames illustrate the fabrication of FG hydrogel thin films using a commercial high-throughput plate and liquid-handling robot. Sequential frames of a zoom on a single well are displayed on top. Time is displayed as hh:mm:ss:ff, where ff indicates the frame number. **d** 2D reconstruction of hydrogel profile embedding fluorescent beads (in magenta) and with HaCaT cells (in gray – RFP-LifeAct™) seeded on top. The z-stack of the hydrogel volume was acquired using confocal imaging. A max intensity projection was applied to the orthogonal views. XY plane scale bar: 200 µm. Orthogonal views (XZ, YZ) scale bars: 20 µm. **e** Assessment of thickness at three different concentrations of FG (5% - orange squares, 10% - purple dots, 20% w/v - black diamonds) in the hydrated state. The data corresponds to a single experiment (3 concentrations, *n* = 5 samples per concentration). For each sample, five fields of view (FOVs) were imaged on the top surface to measure the z-position and calculate thickness. The mean thickness per sample is reported, while flatness is reflected by the consistency of thickness values across the five fields of view. Data are presented as mean ± s.e.m.

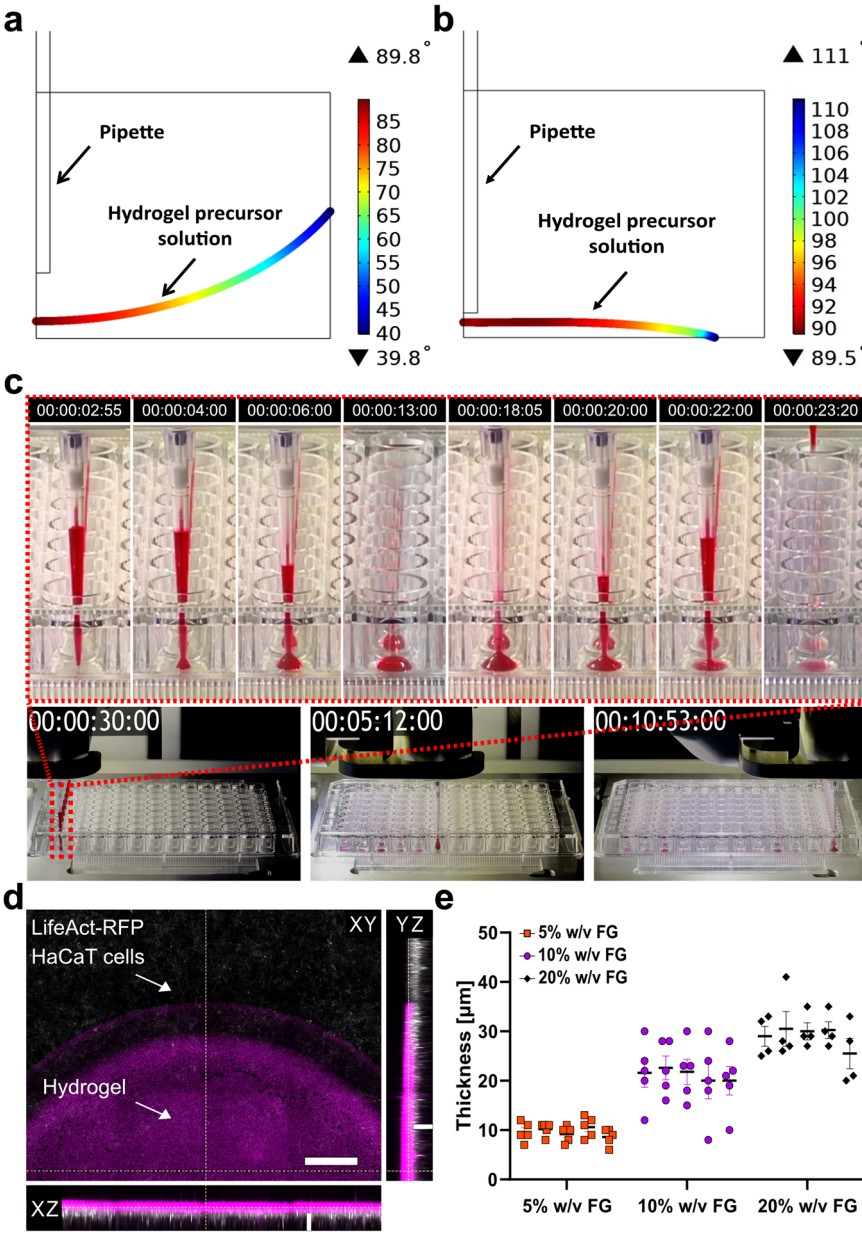

imaging experiments. Digital holography experiments were conducted inside a cell culture incubator for 48 h.

We explored the range of drug concentrations known to affect the HaCaT cell cycle[54,55]. Representative holographic images are shown in Fig. 4 and Figs. S6-S7: vehicle controls (0.1% DMSO at 0 h and 48 h, Fig. 4a–b, Supplementary Fig. S7), positive controls (Geneticin, Supplementary Figs. S6a-b, and S7), nocodazole-treated cells (50 ng mL-1, Fig. 4c), and paclitaxel-treated cells (Supplementary Fig. S6c). Specifically, we tested three concentrations of nocodazole (12.5, 25, and 50 ng mL-1, Fig. 4c and Supplementary Fig. S8a) and three concentrations of paclitaxel (0.5, 2.5, and 12.5 ng mL-1, Supplementary Figs. S6c, S8b). In duplicate runs, cells were seeded either on FG gels or directly on plastic. We successfully imaged HYDRA-prepared gels using digital holography, capturing cell presence and morphology (Fig. 4a–c; Supplementary Figs. S6–S9; Supplementary Movies S5–S6), extending phase microscopy to HTS formats[57].

Moreover, since we imaged all wells every hour for 48 h, we analyzed time-lapse images to measure the total area covered by cells, referred to as "% confluency." This parameter is important because proliferating cells increase their confluency over time, and compounds expected to arrest the cell cycle would ideally delay or arrest this increase (Fig. 4d, Supplementary Fig. S6d, S7a, and S9d-e).

After 48 h, nocodazole-treated cells showed distinct proliferation patterns on hydrogels versus plastic (Fig. 4e). Paclitaxel yielded similar trends (Supplementary Fig. S6e), and higher doses consistently reduced confluency across both substrates (Supplementary Fig. S7b). In these scenarios, the final cell confluences on the hydrogels were twice those observed on the plastic, highlighting the importance of more biomimetic substrates. We also quantified $IC_{50}$ values in the case of plastic or hydrogel substrate: about 17 ng mL-1 on plastic and 18 ng mL-1 on hydrogel for nocodazole, 1.6 ng mL-1 on plastic and 1.8 ng mL-1 on hydrogel for paclitaxel (Fig. 4f and Supplementary Fig. S6f). While yielding similar $IC_{50}$ values, the proliferation kinetics of HaCaT cells on hydrogel or plastic differed, which is also in good agreement with the literature[25]. Together, these results support the notion that HYDRA HTS platforms can provide more biomimetic support for cell proliferation and can be integrated with advanced imaging capabilities and standard analysis pipelines for drug testing applications.

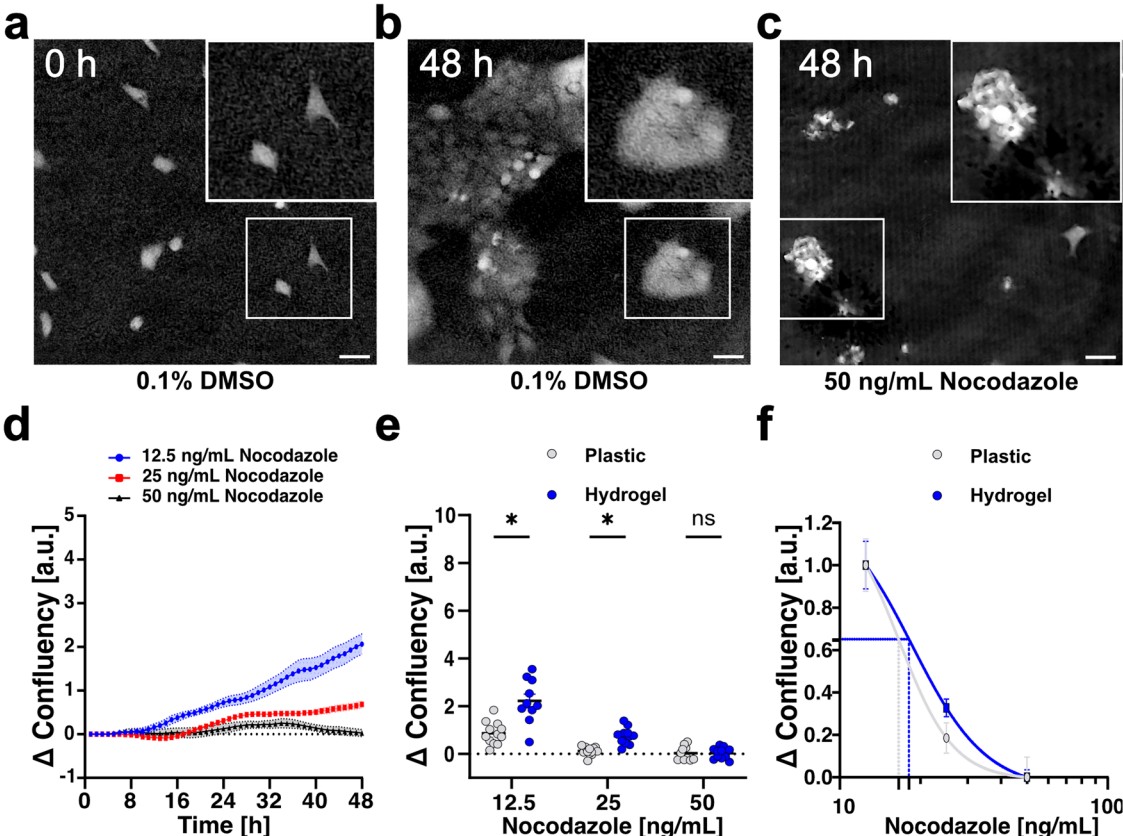

**Fig. 4 | Drug test using phase holographic imaging. a** Holographic image of HaCaT cultured in 0.1% DMSO – vehicle control – after seeding and (**b**) 48 hours after seeding. **c** Holographic image of HaCaT treated with 50 ng mL-1 of nocodazole drug 48 hours after seeding. Scale bars: 50 μm. **d** HaCaT cell confluency vs. time. Cells were cultured in nocodazole-based cell culture media (12.5 ng mL-1 – blue dots, 25 ng mL-1 – red squares, 50 ng mL-1 black triangles) for 48 hours on hydrogel substrates. Data were normalized concerning the initial confluency value. Solid lines represent mean values, and shaded areas represent s.e.m ($n = 12$). **e** Final cell confluency vs. nocodazole concentration (12.5 ng mL-1, 25 ng mL-1, 50 ng mL-1) on plastic and hydrogel substrates. Data were normalized concerning the initial confluency value. (*) stands for significative difference. **f** Nocodazole dose-response curves on plastic ($IC_{50}$, 16.6 ng mL-1) and hydrogel ($IC_{50}$, 18.1 ng mL-1) substrates ($n = 12$). To facilitate visualization of the half-maximal inhibitory concentration (IC50), horizontal and vertical dotted lines mark the intercepts on the Y- and X-axes, respectively, corresponding to the IC50 values for each condition. Data were normalized concerning the initial confluency value ($n = 12$). Image brightness and contrast were adjusted for printed visibility. Data are presented as mean ± s.e.m.

## Drug testing assay using standard fluorescence static imaging

Next, we sought to validate the applicability of HYDRA plates to fluorescence microscopy and to confirm that the digital holography results obtained were consistent with the known mechanisms of action of the drugs used. Nocodazole and paclitaxel influence cell cycle progression by modulating microtubule dynamics, arresting the cell cycle before division (M phase), and eventually leading to cell death by apoptosis[58–60]. To examine the effects of these drugs on the cell cycle, we used HaCaT cells genetically engineered to express fluorescent sensors for the cytoskeletal proteins actin and tubulin as well as the cell cycle sensor FUCCIplex (Fig. 5a)[53]. This cell cycle sensor allowed us to determine the cell phases (Fig. 5b i-v).

In this set of experiments, we used fluorescence microscopy in a temperature- and $CO_2$-controlled incubator to repeat the same drug test performed using holographic imaging. Specifically, we used HYDRA to cast gelatin hydrogels onto plastic 96-well plates. We then counted M-phase-arrested cells on hydrogels or plastic in the presence of the two compounds 24 and 48 h after administration (Fig. 5b–iii, Supplementary Fig. S10–iii, and S11). First, we confirmed that fluorescence imaging is compatible with HYDRA-cast hydrogels, as shown by holographic imaging in the preceding section. Second, we examined evidence of drug effects.

Cells treated with the highest nocodazole concentration displayed nuclear fragmentation (Fig. 5b iv), typically associated with cellular damage before apoptosis[61]. For this reason, cells with a nuclear area smaller than the mean nuclear area of control cells were excluded from the analysis. By

counting the fraction of cells undergoing cell division, we observed the expected dose-dependent accumulation of arrested cells in the nocodazole-treated group (Fig. 5b - v)[54,55]. Furthermore, we observed nuclear fragmentation (Supplementary Fig. S10–iv), accounting on average for ~10–12% of the total cell population across both treatments (Supplementary Fig. S11), and a comparable trend in the number of cells undergoing mitosis in paclitaxel-treated samples (Supplementary Fig. S10–v)[61]. In addition to drug-induced effects, we investigated whether substrate properties influenced single-cell morphology under control conditions (0.1% DMSO). By quantifying projected cell area 48 h after seeding, we found that cells cultured on plastic exhibited significantly larger nuclear areas in G1 and S/G2 compared to those on hydrogels ($p < 0.05$), whereas no significant differences were detected in EG1 and T (Supplementary Fig. S11). These findings are consistent with the established principle that softer substrates restrict cell spreading relative to stiffer ones[62], supporting the view that hydrogel mechanics can modulate cell morphology in a cell-cycle dependent manner.

## HYDRA compatibility with very high throughput formats

Finally, we investigated whether the HYDRA method could be scaled to other high-throughput formats, such as a 384-well plate, and higher-resolution imaging modalities using plates with a high-refractive-index cyclic olefin copolymer (COC) backing. With a working volume of 1 μL, HYDRA prevented meniscus formation by avoiding contact with the walls of each well. With such a small volume, we used 10% w/v FG and 0.5% w/v

**Fig. 5 | Drug test using fluorescence imaging on HYDRA hydrogels cast on inexpensive traditional tissue culture plastic.** This is a worst-case configuration for imaging because photons must travel through a thick polystyrene layer in addition to the thin hydrogel layer (see Fig. 6 for a best-case scenario). **a** Schematic illustration of cell-substrate interaction during the cell cycle. A cell in i) G1 or in ii) S/G2 phase exhibits a degree of spreading based on the substrate. During iii) the mitosis (M) phase, the phase before cell division, the cell undergoes rapid internal architectural changes, assuming the characteristic spherical shape associated with mitotic cells. **b** Static widefield images of HaCaT (RFP – LifeAct™ in gray, GFP – tagged tubulin in green) 48 hours after seeding on gels cast on tissue culture plastic. Cells were treated with 50 ng mL-1 nocodazole. Arrows stand for (i) cells in the G1 phase (in cyan), (ii) cells in the S/G2 phase (in magenta), and (iii) cells in the mitotic (M) phase. Asterisks stand for (iv) nuclear fragmentation. (v) Fold change in M phase cell number after 48 hours vs. nocodazole concentration on plastic and hydrogel substrates. Cell number in the M phase was counted as a fraction of the total cell number in a FOV and then normalized to the vehicle negative control ($n = 12$). Data are displayed as mean ± s.e.m. (*) stands for statistical difference. Images were taken on hydrogel thin layers cast on plastic plates. Scale bar: 25 μm.

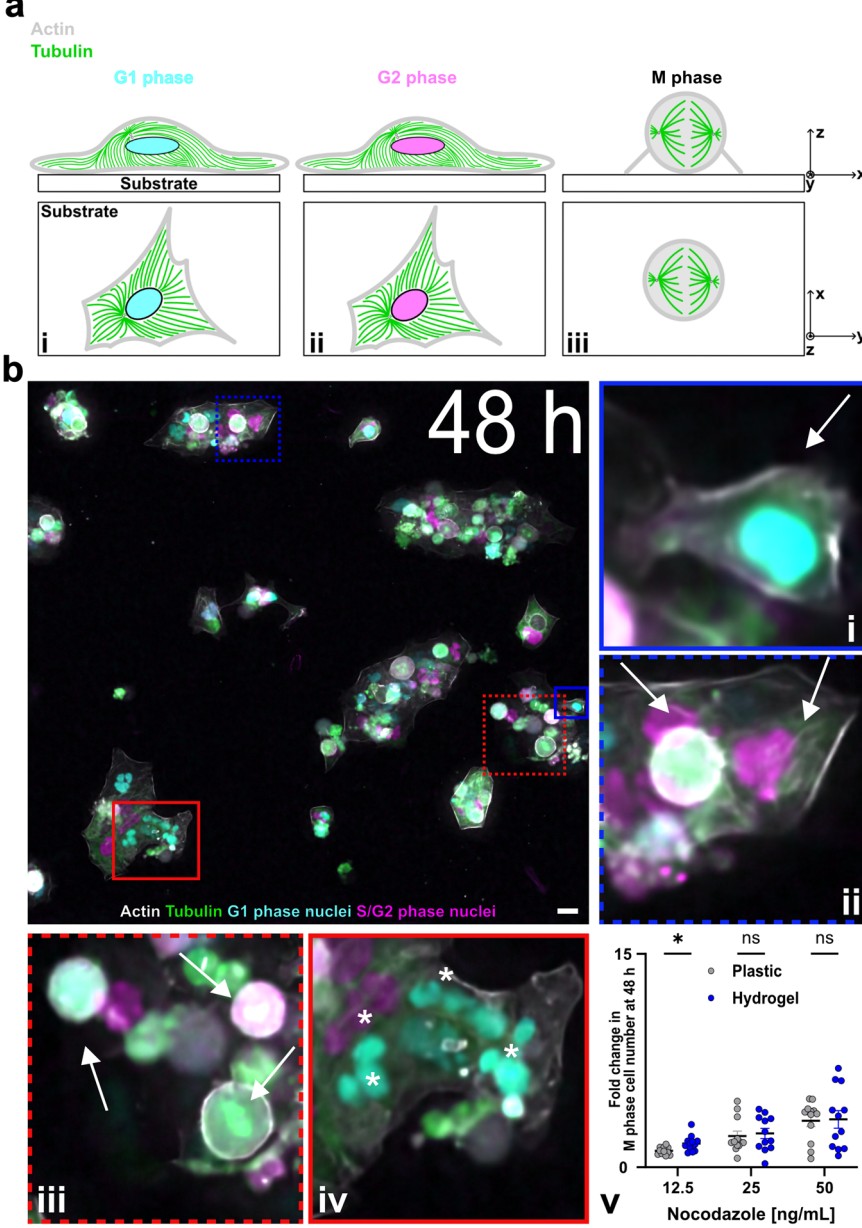

TG to reduce precursor viscosity and optimize dispensing so that the entire plate could be completed in approximately 10 minutes. Given the large number of prepared wells, we developed an automated quality control procedure for HYDRA plates that acquired and analyzed all 384 wells in less than two hours (Fig. 6a and Supplementary Movies S7 and S8). First, we took fluorescence images of the embedded beads with a 4× objective (NA 0.13) in the confocal mode. Second, we calculated the maximum intensity projections to determine whether the gel touched the walls (red box) or if the intensities were inhomogeneous (yellow box). Our quality control revealed a yield of ~90% (green box) with only 6% unusable meniscus profiles (Fig. 6a, red box) and partially usable concave gels (yellow box) (Fig. 6a). We further characterized a sample of inhomogeneous gels with higher-resolution Z-stacks to confirm their non-planar profiles (Supplementary Fig. S12a-b). In contrast, 1536-well plates yielded <10% usable gels, due to robotic limitations in volume handling and positioning accuracy (Supplementary Fig. S12c).

Finally, we demonstrated how HYDRA 384 well plates can be used in advanced fluorescence microscopy applications as required by modern imaging-based HTS approaches (Fig. 6 b)[63]. First, we

performed live-cell imaging experiments by following HaCaT cell proliferation for 18 hours on multiple fields-of-view using a 40× silicon oil objective (NA 1.25) and a perfect focus system (Fig. 6c and Supplementary Movie S9). The image quality was high enough to enable semi-automated segmentation and tracking with modern AI-based image analysis software, such as Stardist and TrackMate (Fig. 6c)[64,65]. Finally, to show that the gels did not prevent the assessment of fine ultrastructural details, we performed high-resolution confocal acquisitions with a 100× silicon oil objective (NA 1.35) (Fig. 6d). In the same field of view, we could observe how microtubule organization changes from a mostly planar network in adhered and spread-out cells (Fig. 6d–i) to a mitotic spindle during cell division (Fig. 6d - ii). Together, our results show that HYDRA can be used in various well-plate formats, from low throughput to HTS, and substrates, from glass to plastic, and that hydrogel cast with this method can be stored at 4 °C for up to two months while retaining their ability to support cell adhesion (Supplementary Fig. S13). In all of these scenarios, HYDRA enables HTS with multiple imaging modalities.

**Fig. 6 | HYDRA-like hydrogels on an imaging-grade 384-well plate. a** Hydrogels were cast in a 384-well plate by using 10% w/v FG/0.5% w/v TG. Hydrogel precursor solutions were mixed with fluorescent beads (shown in magenta) to visualize the gel volume. An automatic quality control analysis was performed to evaluate hydrogel morphology and check if the sample touched the well walls. 91% of the samples resulted in well-deposited planar hydrogels (green boxes), 6% of them formed a meniscus due to touching the walls (red boxes), while 3% of samples appeared with a concave shape (yellow boxes). "x" symbols on well shows wrong classification done by the automatic analysis. Well width: 3.7 mm. **b** 3D rendering of HaCaT cells (actin in gray) on the hydrogel surface (fluorescent beads in magenta). (**c**) 18-hour live fluorescence imaging experiment of HaCaT cells on hydrogels. HaCaT cells express fluorescent sensors for actin (in gray) and the cell cycle (cyan nuclei in G1 phase in cyan, nuclei in S/G2/M phase in magenta). Segmentation masks for selected cells and tracks backward in time are shown. Mask outlines were colored manually on TrackMate according to the cell cycle phase. A tracking diagram is shown where each spot represents a cell at a specific phase of the cell cycle at each time point. The dashed white box indicates the tracking lineage of the highlighted cells. **d** Max intensity projections of the XY, XZ, and YZ planes from a z-stack showing a cluster of cells on hydrogels (actin in gray, tubulin in green, nuclei in cyan and magenta). (**i**) The red square box zooms in on three cells undergoing mitosis, evidenced by mitotic spindles in green. (**ii**) The white dashed box shows the actin and tubulin structures of cells well spread on the hydrogel surface. Zooms were adjusted by applying a Sharpening filter in FIJI[30]. Scale bars: 25 µm.

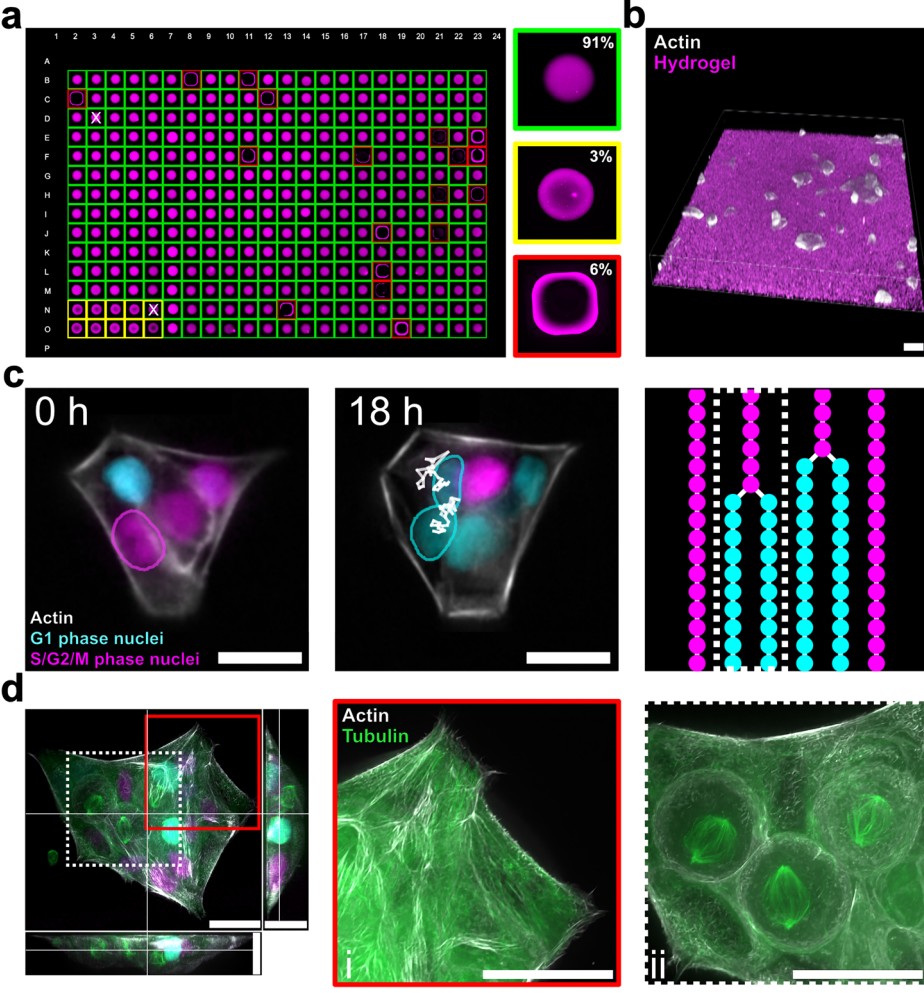

## Discussion

In this study, we asked, What would it look like if it were simple to have biomimetic elements in HTS, specifically tailored to the requirements of imaging-based drug screening assays? Existing solutions for biomimetic functionalization of multi-well plates either yield layers too thin to mask substrate stiffness or too thick for high-resolution imaging[28,30]. Our answer is HYDRA, which is a scalable and automated approach for fabricating thin hydrogel films at the bottom of commercial HTS plates. To do so, throughout this study, we sought material composition and fabrication strategies that would be simple to implement not only in academic research frameworks but also in industry-level scenarios. In this study, we illustrate this dual accessibility by first implementing HYDRA on an entry-level OT-2 platform (Opentrons), which demonstrates feasibility in academic labs with minimal infrastructural burden. To address higher-throughput formats, we then employed the INTEGRA Assist Plus, a platform widely adopted in biotech settings. This progression highlights how HYDRA can bridge academic proof-of-concept studies and industry-scale screening pipelines. Similarly, environmental variables, such as temperature and humidity, can affect gel casting[66–68], especially at the low volumes used for HYDRA fabrication. A systematic comparison of liquid-handling automation and environmental control systems, ranging from other open-source to industry-grade, represents an important future direction that will need industry collaboration.

We selected gelatin for its mild surfactant properties, which enable contact-line pinning and allow excess precursor to be removed during re-aspiration, yielding planar, micrometric-thick hydrogels[69]. We specifically chose cold FG (Fig. 2a–f) instead of the more commonly used mammalian gelatins. Mammalian gelatin is commonly used in manual casting or 3D printing because of its viscous nature at RT, which can be further tuned with a temperature-controlled printhead[40]. While we could have adopted a similar strategy, we preferred standard liquid-handling equipment and sought a gelatin composition that would remain liquid at RT. Although fish and mammalian collagens have several differences in their tertiary and quaternary structures, their denatured products, gelatins, are much more similar[40,41]. Our rheological studies demonstrated that the viscosity of FG at RT could be effectively handled using air displacement pipettes, even at elevated protein concentrations (Fig. 2b). Moreover, the dispensed layer can be effectively crosslinked by microbial TG within a physiological range of stiffness (Fig. 2e, f). This suggests that this strategy can be used as is in applications with a desired cell-adhesive hydrogel. However, our method could be extended to other materials with liquid hydrogel precursors (also photocrosslinkable) to adjust the hydrogel stiffness and cell-binding properties, such as hyaluronic acid, alginate, or methacryloyl FG[70–72]. Importantly, the rheological trends we observed for FG, namely, the increase in viscosity with concentration and its fluidity at room temperature, are in good agreement with established reports on cold-water FG systems, which consistently describe lower gelation and melting points compared to mammalian counterparts[39,40].

We approached the fabrication process similarly: What could be the easiest way to cast hydrogels in HTS plates for the pharmaceutical industry? The HYDRA manufacturing process is simple and involves the dispensing of a small volume of the hydrogel precursor solution at the center of each well, followed by re-aspiration of the same liquid volume (Fig. 1a, b). After dispensing the liquid and wetting the well bottom, a small amount of the

hydrogel precursor remained adhered to the glass/plastic, ensuring that a thin hydrogel layer could be crosslinked in place. Here, the HYDRA method was demonstrated by considering the wells of standard 96- (Fig. 3) and 384-well plates (Fig. 6) with minimal hydrogel modifications. 1536-well formats should also be feasible with more precise robotics that prevent wall contact and retain a boundary layer after reaspiration.

Next, we sought to validate HYDRA in challenging applications from a microscopy standpoint. We reasoned that introducing a hydrogel layer could impact imaging performance and placed ourselves in the most pessimistic scenario by casting the gelatin hydrogel in a plastic 96-well plate. Even under these conditions, we demonstrated the suitability of HYDRA HTS plates for digital holography (Fig. 4) and fluorescence microscopy (Fig. 5) on human epithelial cells engineered with various fluorescent sensors and treated with two drugs. Finally, we used imaging-grade 384-well plates with COC backing to perform advanced live-cell and high-resolution confocal imaging of cells cultured on HYDRA gels (Fig. 6c, d). The extra layer at the bottom of the plate did not affect the standard imaging instruments, and the data quality was sufficient to automate data analysis in ways similar to current state-of-the-art Cell Painting assays and image analysis pipelines[23,24]. The adhesion and proliferation of engineered HaCaT cells that we observed on HYDRA gels are consistent with the well-established behavior of this cell line, which is known to maintain strong adhesion, proliferative capacity, and compatibility with imaging across a range of substrates[53]. In this proof-of-concept study, the goal was to demonstrate feasibility rather than functional re-ranking of drug potency. We intentionally selected HaCaT keratinocytes for their stable cell-cycle reporters to establish compatibility of imaging-based pharmacological assays on thin, mechanically defined hydrogels. The $IC_{50}$ differences between plastic and hydrogel substrates were modest (1.6 vs. 1.8 nM), but consistent with prior reports in similar epithelial systems (e.g., HeLa cells: 2.3 nM at 1 kPa vs. 1.9 nM on glass[73]), while other lines show larger or negligible effects depending on stiffness[74]. These results demonstrate that HYDRA preserves canonical drug responsiveness while providing a reproducible, biomimetic substrate for high-content pharmacological imaging. Future work will extend this framework to stiffness-sensitive models such as hiPSC-derived cardiomyocytes[75,76], to quantitatively assess whether the biomimetic mechanical environment provided by HYDRA enhances predictive power in large-scale pharmacological screens.

The main limitation of HYDRA is the residual glass/plastic rim at the edge of the gel. This strong edge effect can easily be addressed in imaging experiments by acquiring data only from the gel area. However, the presence of cells on the glass/plastic rim remains problematic when extracting DNA, RNA, or proteins from the well, as the cells outside the gel may have properties distinct from those on the gel. Compared to the layer-by-layer and photocrosslinking approaches for obtaining hydrogels in HTS plates, we traded good coverage in favor of physiologically relevant thickness and high-res imaging compatibility[28,30]. Because the final plate remains optically and chemically addressable, combining these and other approaches to generate full-well, more complex microenvironments in future versions of HYDRA will be possible[70,71,77]. Furthermore, future refinements of the finite-element framework could incorporate flow-dependent physics, such as extrusion speed and shear-rate–dependent viscosity, similar to what has been recently explored in extrusion-based bioprinting models[78], thereby extending the predictive power of this HYDRA computational model beyond volume-driven meniscus dynamics.

## Conclusions
We developed HYDRA, a fully automated protocol for fabricating thin planar hydrogels in standard HTS well plates. We used microbial transglutaminase-crosslinked fish gelatin as a cost-effective solution to create tissue-mimicking environments with tunable gel properties and high compatibility with automated liquid-handling. We obtained planar gel profiles by exploiting the contact angle hysteresis during liquid re-aspiration and surface dewetting. We validated this approach using imaging technologies and bioimage analyses that are relevant to HTS. Our approach can

be easily integrated into a standard pharmaceutical pipeline as it uses nothing but readily available instruments and reagents in their expected context of use. Therefore, HYDRA has the potential to improve the predictivity of preclinical trials for drug R&D by providing better mimicking substrates with a cost-effective and HTS-ready solution.

## Methods
### Preparation of transglutaminase-crosslinked fish gelatin hydrogels
Gelatin from cold water fish skin (FG) (G7041-500G, Merck) was dissolved in Dulbecco's phosphate-buffered saline (PBS) without $Ca^{2+}$ and $Mg^{2+}$ (referred to as PBS -/-, C-40232, PromoCell) at 50 °C under magnetic stirring (700 rpm) for 30 minutes and filtered through a 0.45 µm filter (514-1265, VWR). Transglutaminase (TG) powder (Activa® TI Transglutaminase, 1002, Modernist Pantry) was dissolved in PBS -/-, vortexed (700-800 rpm) for 10 minutes, and then filtered through a 0.45 µm filter. FG and TG solutions were used fresh or stored at 4 °C for up to one week. To explore the thickness and flatness of robot-cast hydrogels, samples were prepared with a gelatin content of 0.2 g mL-1 (20% w/v), 0.1 g mL-1 (10% w/v), and 0.05 g mL-1 (5% w/v) while keeping a constant TG concentration of 0.02 g mL-1 (2% w/v). For 5% and 10% gels, 1:1 mixes of 10% and 20% FG stock solutions with 4% TG were used, respectively. A 4:1 mix of 25% FG and 10% TG yielded 20% gels. After mixing, the resulting hydrogel precursor solution was placed in a cell incubator (37 °C, 90% relative humidity, 5% $CO_2$) overnight for crosslinking. During crosslinking in the incubator, a water reservoir and the plate were placed in a sealed box to maintain relative humidity control. Before cell culture, hydrogels were sterilized for one hour using a cell culture hood's ultraviolet (UV) light, then immersed in PBS -/- for at least 6 h to allow for swelling, and rinsed three times in PBS -/- to remove transglutaminase and ammonia byproducts. To cast the hydrogels in 384- (781866, GREINER) and 1536-well plates (4561, Corning), a precursor solution containing 10% w/v FG and 0.5% w/v TG was prepared.

### Rheological characterization of hydrogel viscosity and stiffness
Gelation of FG was assessed by tube inversion using 5, 10, and 20% (w/v) FG solutions in PBS -/- pre-mixed with TG[41]. Samples ($n = 3$) were incubated at room temperature or 4 °C for 72 h and tube inversion was recorded at 24, 48, and 72 h. Rheological measurements were performed using a Discovery HR-2 rheometer (TA Instruments) with a parallel-plate geometry and a Peltier solvent trap to prevent water evaporation. To verify compatibility with automated pipetting, the viscosity of the hydrogel precursors (5-20% w/v) was characterized by flow sweep measurements at room temperature by pipetting 150 µL of FG solution directly onto the rheometer plate. To assess the shear modulus (stiffness), samples were first crosslinked overnight at 37 °C. Hydrogels (≈2 mm high, 2 cm diameter) with varying FG concentrations were cast on a glass slide and formed into cylinders using glass coverslips as lateral spacers and a passivated coverslip on top. Time sweep measurements were carried out at constant strain (1%) and frequency (1 Hz) at 37 °C. Dynamic amplitude measurements were performed at a constant frequency (1 Hz), at increasing strain (0.1–1000%), at 37 °C. Gelation measurements were performed on hydrogel precursor formulations by pipetting 150 µL of solution onto the plate (strain 1%, frequency 1 Hz).

### Computational Fluid Dynamics Simulations
COMSOL Multiphysics (version 6.1, COMSOL) was used to explore the possibility of producing planar 2.5D hydrogels by simulating the dispensing and aspiration processes using a pipette tip in the center of a cylindrical well. The "Laminar Two-Phase Flow, Phase Field" interface (including gravity contribution) was used to track the air-liquid interface. Exploiting the axial symmetry of the system (pipette at the center of a well), a 2D axisymmetric geometry was chosen to save computational power and time. The well was modeled as a rectangle of dimensions Rw and Hw (Supplementary Table S2), with "wetted wall boundary" conditions imposed for the bottom and sidewall boundaries of the well and "open boundary" conditions at the

top of the well, as the upper part of the well is open. In the "wetted wall" condition, the contact angle hysteresis was modeled using advancing and receding angle values reported in the literature to capture the surfactant properties of gelatin and the resulting contact angle hysteresis (Supplementary Table S2)[72]. Specifically, the angle at the wetted wall was set as "advancing" when the contact line velocity, the velocity at which the fluid moves at the bottom of the well, was positive and as "receding" when it was negative. The pipette was modeled as a rectangle ($Rp$ and $Hp$, Supplementary Table S2), with an "interior wetted wall" condition at the lateral wall of the pipette inside the well domain. An "inlet" condition was applied at the upper edge of the pipette to simulate dispensing and aspiration by using positive and negative "fully developed flow," respectively. Material properties were set using the COMSOL library for air and water, and were assigned to the well and pipette domains, respectively. To model the hydrogel precursor solution at the selected COMSOL interface, the viscosity of the liquid phase was set to the value for the 10% w/v hydrogel precursor solution found in this study. The density and surface tension were set at the nominal value for water and the value found in the literature for FG[69]. The mesh was created using the "physics-controlled" option in COMSOL. A mesh convergence analysis based on the meniscus height at the wall confirmed that both "fine" and "extra fine" meshes yielded convergent solutions (<5% error), and therefore, the "fine" option was selected for all reported simulations. Gel flatness was assessed by evaluating the angle at every profile point, which was directly calculated using COMSOL software. The air-liquid interface was considered "planar" when the angle concerning the pipette axis was 90° ± 3°. Flatness was calculated as the percentage of the length of the "planar" air-liquid interface over the well radius. Wall wetting was evaluated via a parametric sweep varying the dispensed volume by ±20% and reducing the well radius by 100 or 200 μm to account for plate tolerance and robot calibration errors.

### Robotic hydrogel fabrication

An open-source, entry-level pipetting system (OT-2 robot, Opentrons) was used to demonstrate proof-of-concept fabrication of micrometer-thick planar hydrogels in formats up to 96-well plates. This platform was deliberately chosen to illustrate HYDRA's accessibility for academic laboratories with minimal infrastructural burden. Specifically, a liquid-handling robot pipeline was implemented using the Opentrons Protocol Designer (Version 7.0.0, Opentrons) to cast and re-aspirate the hydrogel precursor solutions, leaving planar cylindrical hydrogels on the plastic substrates. In terms of hardware, a removable P300 single-channel Gen2 pipette (in short P300) and a P20 single-channel Gen2 pipette (in short P20), an Eppendorf vial rack for the stock solutions (Opentrons), and 8-well plates (μ-Slide 8 Well High ibiTreat, 80806, Ibidi) as a receiving plastic substrate were used. Before casting, the system was calibrated to account for pipette lengths and hardware positions. During casting, the FG and TG stock solutions were mixed into a new vial using P300, using individual tips. The hydrogel solution was then cast into an 8-well plate using an individual P20 tip for each gel. Specifically, the pipette was programmed to dispense 12 μL of liquid at 500 μm from the well bottom with a flow rate of 0.5 μL s−1 and aspirate the same volume at 100 μm from the well bottom with a flow rate of 0.1 μL s−1. To test HYDRA in low-throughput plates, a custom protocol in Python was developed using P300 and P20 pipettes, an Eppendorf tube rack for the stock solutions (Opentrons), and 12-well plates (11012, BIOFIL) as the plastic substrate. Initially, the precursor solutions were pre-mixed in a new tube rack using a P300 pipette. HYDRA-like hydrogels were cast in the first half of a 12-well plate using a volume of 200 μL each. After re-mixing the precursor solutions, a P20 pipette was used to cast multiple 20 μL gels in each well of the second half of the plate. To visualize the final gel, 18 μL was aspirated from each gel.

### Scalable hydrogel fabrication using a multichannel system

To enable scalable hydrogel production in high-throughput formats, a commercial robotic system (Assist Plus, INTEGRA Biosciences) was employed. This platform, designed for high-throughput applications, including biotech settings, was applied to fabricate hydrogels in 96-well and 384-well plates with microliter-scale precision. FG hydrogels were fabricated in 96-well plates using an Assist Plus robot with a 3-position deck (4520, Integra Biosciences). Two 8-channel pipettes were used, namely 300 μL and 12.5 μL 8-channel VOYAGER pipettes (respectively, 4723 and 4721, Integra Biosciences). Two sequential protocols were developed using the VIALAB software (Version 3.1.2.0, Integra Biosciences). First, FG and TG solutions were mixed and dispensed into an 8-row set of Eppendorf vials using the 300 μL pipette. The pipette was then replaced manually with the 12.5 μL model to dispense 12 μL of precursor solution into a 96-well plate (83.3924, Sarstedt), followed by re-aspiration in one column at a time. A premixing step was included before dispensing. Two additional 96-well plates (89626, IBIDI) were prepared similarly to assess long-term hydrogel stability. After crosslinking, the plates were rehydrated in PBS, sealed with parafilm (PM-992, Bemis), and stored at 4 °C for up to two months. Before use, PBS was replaced with cell culture medium, and cells were seeded, confirming that the gels retained their ability to support cell adhesion after storage. Engineered HaCaT cells expressing a fluorescent marker for actin were seeded on the gels to assess hydrogel degradation after one and two months. Static widefield images were captured using a Crest V3 X-Light spinning disk confocal microscope (Nikon) equipped with a Celesta Light Engine source (TSX5030FV, Lumencor), Photometrics Kinetix Scientific CMOS camera, and an air 10× objective (NA 0.30) (MRH00105, Nikon). During acquisition, a Lumencor CELESTA Light Engine source was used to illuminate the sample through the 10× objective at a wavelength of 546 nm for 200 ms with a laser power of 16 ± 2 mW. One field of view per well was automatically acquired using the JOBS module in NIS-Elements[79].

For higher-density formats, a 16-channel VIAFLO pipette (12.5 μL, 4641, Integra Biosciences) was used to dispense hydrogels into 384-well (781866, Greiner; 4681, Corning) and 1536-well plates (4561, Corning). A 10% FG / 0.5% TG solution was pre-mixed in a 25 mL reservoir (4310, Integra Biosciences), then 1 μL and 0.5 μL were dispensed into 384- and 1536-well plates, respectively, followed by a 0.5 μL aspiration. To promote adhesion on glass surfaces, a previously reported silanization protocol for glass coverslips[38] was adapted to 96-well plates with a glass bottom (89627, IBIDI) using an OT-2 robot (Opentrons) under a chemical hood. and Opentrons Protocol Designer (Version 7.0.0, Opentrons). Wells were sequentially treated with 0.1 M NaOH (5 min; 1.09137.1000, Sigma), 0.5% APTMS in ethanol (5 min; 313251000, Thermo Fisher), and 0.5% glutaraldehyde (30 min; 8.20603.0100, Sigma). Static BF images (5 ms exposure) were captured using the same confocal setup with an air 4× objective (NA 0.13, MRH00045) to image the entire gel. Image acquisition was automated via NIS-Elements JOBS[79], and Flat-field correction was applied using the BioVoxel 3D Box plugin in FIJI[64].

### Cell culture

HaCaT cells were genetically engineered to express multiple fluorescent markers, including FUCCIplex (mTurquoise2 and miRFP670) for cell cycle tracking, RFP-LifeAct™ for actin, and EGFP-tagged tubulin for microtubules, following a previously established protocol[53]. The engineered HaCaT cells were cultured in a conventional incubator (37 °C, 90% relative humidity, 5% $CO_2$) using Dulbecco's modified Eagle's medium F-12 Nutrient Mixture (DMEM F-12, 11320-074, Gibco), supplemented with 10% heat-inactivated fetal bovine serum (10270-106, Gibco) and 1% penicillin and streptomycin antibiotic solution (A001-100ML, Himedia). Cultures were maintained below confluency and split at 70% confluence. For all subsequent experiments, cells between passages 50 and 70 (indicated as P50-70) were used.

### Morphological characterization of robot-cast gels in hydrated states

To evaluate the geometry of the hydrogels in the hydrated state, fluorescent beads (excitation 660 nm/emission 680 nm, F8807, Thermo Fisher) were added to the precursor solution at a 50:1 ratio. Hydrogels with 5%, 10%, and 20% w/v gelatin were cast into an 8-well plate (80806, Ibidi), crosslinked

overnight, and rehydrated in PBS -/- before cell seeding. To verify the biocompatibility of the hydrogel and characterize the hydrogel profile by fluorescence confocal microscopy, $5 \times 10^5$ cells mL-1 genome-edited HaCaT cells were seeded. The RFP-LifeAct™ construct (excited at 546 nm) was used for this experiment to visualize actin and obtain fiducial points for assessing the gel profile and plastic substrate. Static confocal images were captured on both the upper and lower surfaces of the gels using a Crest V3 X-Light spinning disk confocal microscope (Nikon) with a Celesta Light Engine source (TSX5030FV, Lumencor), a Photometrics Kinetix Scientific CMOS camera, and an air 20× objective (NA 0.75, MRD00205). The z-level of the cells adhering on top of the gel and of the plastic substrate was recorded to evaluate the gel thickness and flatness.

### Morphological characterization of robot-cast gels in dried states

The radius, thickness, and flatness of the fabricated hydrogels with cylindrical shapes were characterized in dried forms by optical profilometry. Hydrogel samples featuring 5%, 10%, and 20% w/v fish gelatin were cast on glass coverslips (VD12260Y1A.01, Knittel Glasbearbeitungs GmbH) using the scalable protocol previously described and left to crosslink overnight at 37 °C. The samples were then let dry for at least one day at room temperature (24 °C) and atmosphere. The samples in the dried state were then imaged with a Wyko NT9300 white light profilometer (Veeco) using a 20× 0.4 NA CF IC Epi Plan DI Mirau Interferometry Objective (Nikon) in vertical scanning interferometer (VSI) mode. The data was processed, analyzed, and 3D rendered using Vision 4.2 software (Veeco). At least three hydrogels per condition were imaged, and x- and y-cuts were taken in correspondence with the center of each gel to estimate thickness and flatness. Spherical aberration in the optical setup and substrate curvature were corrected using a second-order polynomial normalization using the thickness values of the glass substrate around the hydrogels.

### Drug screening using digital holographic long-term imaging

Nocodazole (M1404-50MG, Merck) and paclitaxel (10461, Cayman Chemical) were chosen as test compounds. Stock solutions (5 mg mL-1 in dimethylsulfoxide, D2650-100ML, Merck) were aliquoted in 1 mL Eppendorf tubes and stored at -20 °C. HaCaT cells (P50-P70, $5 \times 10^4$ cells mL-1) were seeded on 10% w/v FG hydrogels 24 h before treatment. Cells were exposed to three concentrations of each drug: 12.5, 25, and 50ng mL-1 for nocodazole; 0.5, 2.5, and 12.5ng mL-1 for paclitaxel. Controls included 0.1% DMSO (negative) and 1500µg mL-1 Geneticin (10131-027, Gibco; positive), with 12 replicates per group. After adding drugs, the 96-well plate was immediately placed on the motorized stage of a HoloMonitor M4 (Phase Holographic Imaging AB) inside a cell incubator (37 °C, 90% relative humidity, 5% $CO_2$). Using HoloMonitor Appsuite software (Version 4.0.1, Phase Holographic Imaging PHI AB), one field of view per well was imaged every hour for 48 h using an Olympus PLN 20× objective (NA 0.4) (N1215900, Evident) and a low-power laser unit (635 nm, 0.2 mW cm⁻²). Time-lapse images were analyzed in FIJI to calculate cell confluency, defined as the ratio between the area covered by cells and the total field of view.

### Drug screening using static widefield fluorescence imaging

Complementary to the holographic assay, a 96-well platform was used to capture static widefield images of cells (P50-P70, $5 \times 10^4$ cells mL-1) on the gels at 24 and 48 h after drug treatment. One image per well was captured using a Ti2 inverted microscope (Nikon) equipped with an air 20× objective (NA 0.75, MRD00205). Lumencor CELESTA Light Engine with 546 nm (laser power 16 ± 2 mW, exposure 50 ms), 477 nm (28 ± 3 mW, 50 ms), 446 nm (20 ± 2 mW, 10 ms), and 638 nm (37 ± 4 mW, 200 ms) wavelengths was used to image cell actin, tubulin, and nuclei in G1 and S/G2/M, respectively. The semi-automated imaging protocol was implemented using the JOBS module in Nikon NIS-Elements. For each well, the objective was positioned at the center, the focus was manually adjusted at 477 nm, and 4-channel images were recorded. The entire protocol was executed twice: first to capture cells on hydrogel substrates, and then to acquire cells on the plastic rim surrounding the gel.

### Cell confluency analysis using digital holographic long-term imaging

Cell confluency, defined as the percentage of the field of view occupied by cells, was extracted from time-lapse images acquired with the HoloMonitor M4. Datasets were exported from HoloMonitor Appsuite (Version 4.0.1, Phase Holographic Imaging) and processed using FIJI[64]. The images were processed as follows: "Non-local means (NML) denoising" ImageJ plugin (auto-estimated sigma, smoothing factor "1")[80,81], Auto-threshold ("Triangle" method, default settings provided by ImageJ), morphological operators including "Opening," "Closing" and "Filling holes" operators (ImageJ) and Particle analysis ("Analyze Particles" ImageJ plugin) to extract "%Area" values, i.e., cell confluency values. To reduce quantitative phase imaging artifacts, a moving average (window size = 4) was applied to time-series data. For frames exceeding 30% confluency, manual adjustments were required due to the reduced accuracy of automatic segmentation.

### Automated quality control of HYDRA-like hydrogels in HTS plates

Automated acquisition and quality control analyses were performed to evaluate hydrogel fabrication in 384-well (781866, GREINER) and 1536-well (4561, Corning) plates. Using the JOBS module in Nikon NIS-Elements, static widefield z-stacks (30 µm z-step size) were automatically acquired at the center of each well with a Crest V3 X-Light spinning disk confocal microscope (Nikon) equipped with a Celesta Light Engine source (TSX5030FV, Lumencor), a Photometrics Kinetix Scientific CMOS camera, and an air 4× (NA 0.13, MRH00045). Illumination was provided at 546 nm (16 ± 2 mW, 300 ms) and 638 nm (37 ± 4 mW, 30 ms) wavelengths to image well walls and hydrogels embedding fluorescent beads, respectively. A custom FIJI macro script was used to process the 2-channel z-stacks. Briefly, each image was cropped and rotated to center the well within the field of view. The maximum intensity projection was then obtained from the 2-channel stack input. In the well wall channel, gamma correction (0.5), auto-thresholding ("Triangle"), morphological operations (opening, closing, hole filling), outlier removal (radius 10 px), and edge detection were applied. The hydrogel channel underwent similar processing, using the "Default" threshold method. A logical AND operation between the hydrogel and well wall masks identified gels in contact with the well edges ("bad"). A secondary analysis on the hydrogel projection (gamma = 2, auto-threshold, closing) classified gel shape based on mean gray value: ≥250 was defined as "planar," otherwise "concave." Well classifications ("bad," "planar," "concave") were saved in an Excel file. A Python script then tiled the processed images by classification, overlaying each well with a colored frame: green (planar), red (wall contact), or yellow (concave).

### Profile extraction of fish gelatin hydrogels in a 384-well plate

To assess hydrogel profiles in 384-well plates (781866, Greiner), fluorescent beads were embedded in the gel matrix as previously described. After overnight crosslinking and rehydration in PBS -/-, large field-of-view confocal z-stacks were acquired using a Crest V3 X-Light spinning disk microscope (Nikon) equipped with a Celesta Light Engine (TSX5030FV, Lumencor), a Photometrics Kinetix CMOS camera, and a 20× air objective (NA 0.75, MRD00205). A custom Python script was developed to extract profiles from $n = 3$ for concave and $n = 5$ for planar gels. Briefly, z-stack images were processed by applying Otsu thresholding and morphological closing to differentiate the gel from the background. Gel height was determined by the distance between the first and last detected signals, multiplied by a 2 µm z-step size. Profiles were smoothed using a 15-pixel moving average filter (~10 µm, pixel size = 0.67 µm).

### High-resolution live fluorescence imaging

Cells (P50-P70, $1 \times 10^3$ cells mL-1) on gels cast in 384-well plates (4681, Corning) were analyzed using long-term widefield acquisition and static confocal fluorescence imaging. For long-term widefield imaging, one image per well was taken every 15 minutes for 18 h. Three wells per plate were considered. A Ti2 inverted microscope (Nikon) was equipped with a 40× silicon oil objective (NA 1.25) (MRD73400, Nikon). Lumencor

CELESTA Light Engine with 546 nm ($16 \pm 2$ mW, 200 ms), 477 nm ($28 \pm 3$ mW, 30 ms), 446 nm ($20 \pm 2$ mW, 10 ms), and 638 nm ($37 \pm 4$ mW, 60 ms) wavelengths was used to image cell actin, tubulin, and nuclei in G1 and S/G2/M, respectively. For high-resolution static imaging of cell clusters, confocal z-stacks (0.3 μm z-step size) were captured using a Crest V3 X-Light spinning disk confocal microscope (Nikon) equipped with a Celesta Light Engine source (TSX5030FV, Lumencor), Photometrics Kinetix Scientific CMOS camera, and a 40× (NA 1.25, MRD73400) or 100× silicon oil objective (NA 1.35, MRD73950). Z-stacks were processed using AI Denoising and Richardson–Lucy deconvolution (50 iterations) in NIS-Elements software, and visualized as XY maximum intensity projections, orthogonal views (XZ, YZ), or 3D renderings.

## Cell cycle analysis using fluorescence widefield imaging

Cell cycle analysis was conducted on both long-term time-lapse and static widefield fluorescence images. In both cases, nuclei were classified using FUCCI-based markers, as previously reported[53]. Briefly, the cyan (G1) and magenta (S/G2/M) channels were denoised (non-local means[80] or median filtering), combined, and segmented using the pre-trained StarDist model (Versatile fluorescent nuclei)[65]. ROIs were extracted, and cells were classified into early G1 (no signal), G1 (cyan signal above threshold), S/G2/M (magenta signal above threshold), or transition (T; both signals above threshold), following background subtraction and intensity thresholding. Segmentation masks were manually curated when necessary (~10% of frames). For static images, mitotic cells were additionally identified using the tubulin (green) channel. After removing the cyan signal to correct for bleed-through, tubulin intensity was measured within nuclei. The background was estimated from the surrounding cytoplasm. A cell was classified as mitotic if tubulin intensity exceeded background by ≥0.2 (plastic) or ≥0.4 (hydrogel). These predetermined values were established to automate the classification analysis, similar to a manual evaluation from an expert. Notably, the tubulin threshold for cells on hydrogel was doubled in value compared to that of cells on plastic to account for the autofluorescence background from the gel. In time-lapse data, segmented label images were imported into TrackMate[82]. The LAP tracker was used (50 μm max distance, 3-frame gap closing), with nuclear intensity-based penalties. Lineage trees were generated with phase-specific color coding[83]. The linking of daughter cells was performed manually. Manual corrections (~5%) were applied to recover daughter cells after mitosis, when FUCCI signals were briefly absent. Quantitative single-cell morphology was assessed by measuring the projected nuclear area under control conditions (0.1% DMSO) on both plastic and hydrogel substrates, using ROIs extracted from the segmentation; analyses were stratified by cell-cycle phase (EG1, G1, S/G2, T) as identified in the imaging workflow. Under high-concentration drug treatments, cells exhibiting nuclear fragmentation were identified by visual inspection across five representative fields of view, manually counted, and excluded from analysis.

## Statistics and reproducibility

Statistical analyses were performed using GraphPad Prism 10 software (GraphPad Software, USA). Mann–Whitney multiple tests were performed to compare confluency in long-term time series and static images. All results are presented as the mean ± standard error of the mean (s.e.m.). The conceptual drawing of the liquid-handling robot in the table of contents graphic was created in BioRender. (Di Sante, M. (2025) https://BioRender.com/e23i519).

## Reporting summary

Further information on research design is available in the Nature Portfolio Reporting Summary linked to this article.

## Data availability

All relevant data that support the findings of this study are available from the corresponding author upon reasonable request.

## Code availability

All analysis scripts and image processing routines used in this study are available in the public repositories https://github.com/Synthetic-Physiology-Lab/hydra-hts-hydrogel-analysis and https://github.com/Synthetic-Physiology-Lab/fucciphase.

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

## Acknowledgements

The authors thank Prof. Frank Niklaus for granting access to the profilometer and Theocharis-Nikiforos Iordanidis for the support during the characterization procedure. This work was supported by the European Research Council (ERC) Starting Grant No. 852560 to F.S.P. and by the Italian Ministry of Education, University, and Research (MIUR) (FARE2020, Grant No. R20ZE54CTK) to F.S.P.

## Author contributions

E.T. and F.S.P. contributed equally to this work. E.T. and F.S.P. conceived and demonstrated the concept of fabricating HYDRA-fabricated hydrogel thin films in high-throughput plates. E.T. conducted all characterization of the HYDRA method, managed cell handling, performed drug testing and long-term imaging experiments, and analyzed the data. M.D.S. provided the engineered cells used in the experiments. M.M. and J.U.L. conducted rheological measurements of fish gelatin. B.H., A.E., and E.C. designed and conducted the COMSOL simulations for casting gels in high-throughput wells. M.P. wrote the code to program the microscope for automated fluorescence imaging. J.Z. created the Python code to analyze cell cycle phases of fluorescence images. M.D.S., M.P., J.Z., S.G., M.M., J.U.L., F.A., E.C., and A.E. supervised the experiments and contributed to manuscript preparation. E.T., A.E., and F.S.P. co-wrote the initial manuscript. All authors discussed the results and provided feedback on the manuscript.

## Competing interests

The authors declare no competing interests.
