## [Transparent Peer Review file · Communications Engineering]

Fabrication of cell culture hydrogels by robotic liquid handling automation for high-throughput drug testing

Corresponding Author: Professor Francesco Pasqualini

Version 0:

Reviewer comments:

Reviewer #1

(Remarks to the Author)

It is well established that cellular behaviour like, gene expression, drug responsiveness and morphology is strongly influenced by mechanical cues such as substrate stiffness. Therefore, the development of culture substrates that more accurately mimic the in vivo cell environment, while remaining compatible with standard high-throughput liquid handling and imaging systems in a well plate-based cell culture format, represents a promising strategy for enhancing the predictive power of in vitro drug screening assays. The novel approach presented here, including the quality control of the fabricated well plates, appears to be both feasible and practical. However, the proposed system is only suited for applications relying exclusively on imaging-based analysis. This limitation arises from the challenge of separating cells proliferating on the rigid surface adjacent to the hydrogel from the cells on top of the hydrogel when extracting RNA, proteins or retrieving cells for techniques like flow cytometry.

Overall, the described methods are generally comprehensive and appropriately detailed, but it is recommended to also report the cell seeding densities and passage numbers used in the experiments. The manuscript is overall well written and well-structured; however, the following points should be addressed:

o The authors cite references in conjunction with statements that pertain to the results or investigations of the current study. It should be clearly distinguished which findings originate from the present work and which are based on previously published literature (Line 91, Line 131).

o Abbreviations should be introduced before first use (for example Fig. 3: FOV, Line 113: TG).

o Line 152: The authors mention "hydrogels displayed physiologically relevant stiffness". What is regarded as physiological stiffness is highly dependent on the specific tissue/cell type used. Please specify.

o Line 223: "three concentrations of nocodazole (12.5, 25, and 50 ng mL⁻¹, shown in Fig. 4a-b-c)". Figure 4 a and b show holographic images of HaCaT cultured in 0.1% DMSO.

o Figure 3e: How is flatness of FG hydrogels displayed in that figure?

o Line 325: What do the authors mean by "it remains viable for two months"?

o More references should be included to support general statements throughout the manuscript. Where possible, older citations should be replaced or complemented with recent literature (for example: Line 55, Line 195, Line 75-80).

o Line 136: Please correct: "...storage modulus (G') and loss modulus (G'')..."

o Have the authors analysed the swelling behaviour of the hydrogel in contact with culture medium over the full duration of the cell cultivation experiments? Additionally, have the authors investigated how far in advance the well plates can be prepared prior to cell seeding? In particular, how was the reported maximum storage duration of two months (Line 501) determined?

Reviewer #2

(Remarks to the Author)

Introduction

The manuscript presents HYDRA, a novel method for fabricating planar hydrogels in multiwell plates using robotic liquid handling, with the aim of enhancing the physiological relevance of high-throughput screening (HTS). The authors make a case for the need to improve substrate biomimicry in early drug testing pipelines, highlighting limitations in current glass/plastic systems and the impracticality of more complex 3D models like organoids and organs-on-chip at HTS scale. The motivation is real, though the introduction could better acknowledge concurrent innovations in microfabricated 2.5D

systems, which share overlapping goals. The rationale for choosing fish gelatin over more established materials such as PEG or collagen could also benefit from deeper comparative justification early on.

Results

The authors validate HYDRA using rheological assays, finite element modeling, and imaging-based drug testing. The robustness of their mechanical characterization is commendable, particularly their quantitative handling of viscosity and stiffness, and their demonstration of control over hydrogel flatness via dispensing parameters. The biological data, including dose-response assays with nocodazole and paclitaxel, are convincing and confirm that the thin gels support normal cell growth and drug responsiveness. However, the differences in IC50 values between plastic and hydrogel substrates are relatively modest, which raises the question of how much functional advantage HYDRA offers over standard coatings. The authors report increased final confluency on hydrogels, but whether this improvement translates to better predictive power in real-world drug screens remains speculative and unquantified.

Discussion

The discussion is thoughtful and transparent, particularly in acknowledging the edge effects from residual plastic rims and potential limitations in biochemical extractions. The authors are right to emphasize the compatibility of HYDRA with standard HTS infrastructure, which is indeed one of its most attractive features. However, the central claim that HYDRA offers a better balance between biomimicry and scalability needs to be better substantiated. For instance, while flatness is shown to be improved, how this affects downstream image analysis quality or assay reproducibility is not explicitly tested. Similarly, while the gel thickness is in a biologically meaningful range, it's unclear how uniform this remains across large batches or over time? The manuscript would benefit from more quantitative comparison to other thin hydrogel methods, particularly in terms of throughput, cell behavior variability, and imaging performance.

Materials & Methods

The methods section is ok, with protocols and a range of technologies spanning computational modeling, rheology, and automated microscopy. The use of off-the-shelf robotics is a strength, but it's unclear how easily these workflows could be implemented in other labs without detailed calibration. While the authors stress the minimal infrastructural burden, liquid handling at the microliter scale with precise aspiration timing remains non-trivial. Additional benchmarking across different robotic platforms would have strengthened claims of generalizability.

In conclusion, this is a good contribution. The work is technically acceptable and biologically relevant, though some claims regarding its practical impact in HTS could be more rigorously supported.

Reviewer #3

(Remarks to the Author)

In this paper, Torchia et al present a novel method for consistent fabrication of hydrogels in high-throughput screening (HTS) plates using automated liquid handling. Using a combination of high resolution imaging and multiple pharmacological assays performed on both 96 and 384 well plates, they also demonstrate the scalability of their technique and its potential applicability to in-vitro drug testing. Overall, the study is robust and well-planned; however, there are some aspects that needs additional clarification/revision as listed below: -

i) Page 7 line 148- There seems to be a confusion between storage modulus and shear modulus, as the same notation G' is used for both here. Please clarify how the shear modulus was calculated for your analysis in fig 2 e&f, because it is different from storage modulus.

ii) The authors have performed time and amplitude sweep as well as the variation of viscosity with shear rates. Was there a specific reason to not perform frequency sweep analysis?

iii) Page 9- Section 3.2 - The simulation of meniscus formation using FE modeling provides useful insights on how to identify the optimal dispensing volume. However, it may be useful to study the effect of extrusion speeds/ flow rates on meniscus formation as well as the material viscosity is shear rate dependent.

iv) Page 10 - Fig 3e - The gel thickness is higher for increased concentrations of FG. Considering all gels have similar volumes, is this related to gel swelling properties? Some clarifications here may be useful.

v) Page 14, Fig 4f- nocodazole concentrations corresponding to the IC50 value can be shown on the curves if possible.

vi) Page 15- line 275- What would be the approximate % of cells of the total population that exhibited nuclear fragmentation and were excluded from analysis?

vii) It is impressive that the authors have done several high-resolution imaging-based analyses on cells in both 96 and 384 well plates for several drug treatments and compared the results between cells cultured on plastic plates and hydrogels. However, some basic quantification on how much the single cell and nuclear morphologies differ in both cases may be useful to report as well.

viii) Page 26- Line 464 - How did you specifically choose this mesh size? was a mesh convergence analysis performed?

Version 1:

Reviewer comments:

Reviewer #1

(Remarks to the Author)

The authors have adequately addressed all previous concerns in the revised version of the manuscript.

Reviewer #2

(Remarks to the Author)

Response 2.1 is not acceptable. And the references mentioned are too old (2 are from 2012). What was expected was comments/comparisons in regards to commercial bioprinters and their capabilities (e.g., Rastrum from Inventia Life Sciences). Then the 'the rationale for choosing fish gelatin over more established materials such as PEG or collagen' was not addressed at all.

Response 2.2 did not answer the problem.

Response 2.3: The additional figure demonstrates how heterogeneous the system and this non-reproducible. Then the claim that 'iii) a comparison with existing platforms across throughput levels has been added' was meant as an actual comparison, although this is acknowledged that this could be difficult. As least a literature comparison should be provided.

Reviewer #3

(Remarks to the Author)

The authors have sufficiently addressed all my previous concerns and have revised the manuscript accordingly. I have no further comments. Thank you.

We thank the editor and the reviewers for their work improving our manuscript. We present below a detailed response to the received feedback, as well as a summary of the modifications implemented in the manuscript (a revised version was also uploaded).

Editor

Comment 1

You state that Machillot et al.'s layer-by-layer method produced coatings that were insufficiently thick. Why can these layers not be built up to a suitable thickness? Would it simply be too time consuming?

Response 1

We thank the Editor for the opportunity to clarify this point. Machillot et al. deliberately optimized layer-by-layer (LbL) microplate films for thin, optically transparent PEM coatings spanning “a few tens of nanometers to a few micrometers,” using plate tilting and ~10% over-aspiration to ensure uniformity: well-suited to HTS but not to create thicker, soft substrates that shield cells from the underlying rigid substrate. Cells typically sense the underlying rigid substrate unless the compliant layer is tens of micrometers thick for single cells (~10–20 μm) and substantially thicker for colonies (half-max response ~54 μm at 1 kPa). Achieving ≥ 50 –100 μm by LbL in 96/384-well formats would require hundreds to thousands of deposition/rinse cycles with accumulating defect risk; stronger crosslinking (e.g., EDC/NHS) also increases film stiffness, undermining the intended mechanical decoupling. Thus, the constraint is both practical and physical, not merely time.

Modification 1

- **Old text (Line 78-80):** At the other end of the spectrum, Machillot et al. utilized a layer-by-layer coating to create single micron hydrogel layers, but these coatings were insufficiently thick to shield cells from the underlying stiff plastic substrate, influencing cellular responses.
 - **New text (Line 86-94):** *Machillot et al. established an automated LbL process in microplates yielding optically transparent PEM films with thicknesses from a few tens of nanometers to a few micrometers, aided by plate tilting and ~10% over-aspiration to improve uniformity. However, effective mechanical shielding of adherent cells from the rigid polystyrene generally requires soft layers above mechanosensing length-scales, on the order of ~10–20 μm for single cells and considerably thicker for colonies (half-maximal depth response ~54 μm at 1 kPa). Scaling LbL to such thicknesses in 96/384-well formats would entail impractically many deposition/rinse cycles with elevated risk of defects and delamination; moreover, increased chemical crosslinking (e.g., EDC/NHS) stiffens PEMs and counteracts the desired compliant interface.*
-

Comment 2

The introduction, beyond Refs 20-22, doesn't discuss other examples of hydrogels in well plates. For example, Merck/Sigma-Aldrich produces well plates with hydrogels, some advertised as being designed for "High-Throughput 3D Cell Culture" (see links below). As commercial products exist, is it the case that these products/solutions are not appropriate for your use case? Are they not the right thickness or flatness?

- <https://www.sigmaaldrich.com/GB/en/technical-documents/technical-article/cell-culture-and-cell-culture-analysis/3d-cell-culture/high-throughput-truegel-3d-hydrogel-plates?srsId=AfmBOorAEdPOxEvDyR2JBo3U9gVLKBQZHB2NUhsm448OQbglaZt4heOq>
- [https://www.slas-discovery.org/article/S2472-5552\(22\)06981-7/fulltext](https://www.slas-discovery.org/article/S2472-5552(22)06981-7/fulltext)

Response 2

These references describe 96-well plates with pre-assembled PEG-based hydrogels, conceptually similar to IBIDI angiogenesis plates (<https://ibidi.com/content/322-angiogenesis-assays>) in that they use a recessed bottom to allow controlled hydrogel dispensing.

The availability of such commercial systems highlights the strong interest in reliable high-throughput hydrogel platforms. By providing ready-to-use matrices across 96 wells, these plates enable the scaling up of 3D assays and demonstrate how the field is moving toward standardized hydrogel formats for drug screening and phenotypic studies. That said, their design is clearly optimized for 3D applications: the gels are typically about 1 mm thick (on top of ~180 μm glass) and often include stiffness gradients from the substrate toward the bulk to promote invasion and migration. These features are well-suited for volumetric assays; however, they also limit high-resolution imaging to a narrow region near the glass–hydrogel interface.

For our purposes, these systems are not suited to image-based high-throughput drug screenings, as the gel thickness limits access to a flat, optically compatible surface at the gel–medium interface, which is essential for high-resolution imaging. In short, such commercial products address high-throughput 3D screening needs, but not the demand for early high-throughput biomimetic platforms, the gap our work seeks to fill. We clarified this in the introduction.

Modification 2

- **Old text (Line 75-80):** Recent efforts sought to overcome these limitations. Brooks et al. formed PEG-based hydrogels in 96-well plates through photocrosslinking, yielding tunable stiffness, yet the thick (500–1000 μm) gels compromised optical resolution. At the other end of the spectrum, Machillot et al. utilized a layer-by-layer coating to create single micron hydrogel layers, but these coatings were insufficiently thick to shield cells from the underlying stiff plastic substrate, influencing cellular responses.
- **New text (Line 82-99):** *Achieving uniform thin hydrogel layers within multiwell HTS plates remains technically challenging: the meniscus at well edges creates curvature that disrupts cell attachment, homogeneous growth, and high-resolution microscopy (Trask et al., 2018, Nienhaus et al., 2023). PEG and hyaluronic acid–based hydrogels have been fabricated in 96-well plates through photocrosslinking, yielding tunable stiffness and viscoelasticity, yet the thick (500–1000 μm) gels compromised optical resolution (Brooks et al., 2018, Skelton et al., 2024). Machillot et al. established an automated LbL process in microplates yielding optically transparent PEM films with thicknesses from a few tens of nanometers to a few micrometers, aided by plate tilting and ~10% over-aspiration to improve uniformity (Machillot et al., 2010). However,*

effective mechanical shielding of adherent cells from the rigid polystyrene generally requires soft layers above mechanosensing length-scales, on the order of ~10–20 μm for single cells and considerably thicker for colonies (half-maximal depth response ~54 μm at 1 kPa) (Buxboim et al., 2010). Scaling LbL to such thicknesses in 96/384-well formats would entail impractically many deposition/rinse cycles with elevated risk of defects and delamination; moreover, increased chemical crosslinking (e.g., EDC/NHS) stiffens PEMs and counteracts the desired compliant interface. Similarly, commercial hydrogel plates (e.g., TrueGel 3D from Merck/Sigma-Aldrich) provide pre-assembled PEG-based matrices in 96-well format, reflecting the demand for high-throughput culture platforms (Zhang et al., 2017). However, as with Brooks et al., the resulting gels are several hundred micrometers thick, which restricts high-resolution imaging to planes near the well bottom and makes them unsuitable for 2D mechanobiology assays that require a flat, optically accessible surface (Brooks, et al., 2018, Zhang et al., 2017).

New references:

- Nienhaus, F., Piotrowski, T., Nießing, B., König, N. & Schmitt, R. H. Adaptive phase contrast microscopy to compensate for the meniscus effect. Scientific Reports 13, 5785 (2023). DOI: <https://doi.org/10.1038/s41598-023-32917-6>
- Skelton, M. L. et al. Modular Multiwell Viscoelastic Hydrogel Platform for Two- and Three-Dimensional Cell Culture Applications. ACS Biomater. Sci. Eng. 10, 3280–3292 (2024). DOI: <https://doi.org/10.1021/acsbiomaterials.4c00312>
- Zhang, N. et al. Soft Hydrogels Featuring In-Depth Surface Density Gradients for the Simple Establishment of 3D Tissue Models for Screening Applications. SLAS Discovery 22, 635–644 (2017). DOI: <https://doi.org/10.1177/2472555217693191>

Comment 3

You mention that fish gelatine is used due to its mild surfactant properties. As a surfactant would reduce the surface tension, it could reduce the curvature of the meniscus. Did you consider using a different hydrogel material (e.g. agar) and adding surfactant to reduce the curvature of the surface?

Response 3

We would first like to clarify a potential source of misunderstanding. In the manuscript, when referring to the “mild surfactant properties” of fish gelatin, we specifically meant its ability to promote pinning of the droplet’s contact line during dispensing, thereby enabling more uniform spreading, rather than a reduction in meniscus curvature.

At the beginning of this project, we did investigate whether surfactant addition could mitigate meniscus effects. We compared multiple hydrogel precursors (fish gelatin, alginate, agarose) as well as water, both alone and supplemented with commonly reported surfactants (Pluronic 0.1% w/v, urea 3% w/v) (Hancock, M. J., 2012; Mihajlovic, M., 2018). To directly visualize meniscus-induced distortions, we placed a square grid beneath 96-well plates and imaged the wells before and after liquid addition (Figures 1–2). As illustrated in Figure 1 (water was chosen as an example to visualize the overlapped grids), the meniscus acted as a lens that shrinks the underlying grid.

Figure 1. Meniscus distortion in 96-well plates for water and chemicals acting like surfactants. Square grids placed beneath the wells appear distorted due to the lens effect of the curved liquid–air interface. Examples are shown for empty wells, Milli-Q water, fish gelatin (10% w/v), Pluronic (0.1% w/v), and urea (3% w/v). Right panel: overlaid grid outlines before (green) and after (red) liquid addition illustrate the contraction induced by the meniscus.

Figure 2. Comparison of hydrogel materials (alginate 0.5% w/v, agar 0.5% w/v) with and without added surfactants. Wells were imaged over a reference grid to visualize curvature-induced distortions. Supplementation with Pluronic (0.1% w/v) or urea (3% w/v) did not visibly mitigate the lensing effect of the meniscus.

To quantify these distortions, we tracked the displacement of grid nodes and applied affine transformations (translation, rotation, isotropic/anisotropic scaling, shear). This analysis provided metrics such as contraction percentage, anisotropy, and residual non-uniform distortion (Figure 3; data summarized in Table 1). These measurements confirmed that even in the presence of surfactants, the characteristic “lens effect” of the liquid–air interface persisted, and no reproducible reduction in meniscus curvature was observed.

Figure 3. Quantitative analysis of meniscus-induced distortion using affine transformation fitting. Example shown for water: grid nodes were tracked before and after liquid addition (blue = before, orange = after, green = fitted transformation). The affine fit accounts for translation, rotation, scaling, and shear. Residual mismatches indicate non-uniform distortions beyond simple scaling.

	Condition	Mean scale	Sx	Sy	Anisotropy $ s_x - s_y $	Contraction %	RMS residual
0	alginate	0.9	0.9	0.9	0.0	12.3	1.2
1	alginate&urea	0.9	0.8	0.9	0.0	13.8	1.1
2	agar&urea	0.9	0.9	0.9	0.0	14.3	1.1
3	urea	0.8	0.8	0.8	0.0	19.4	1.2
4	alginate&pluronic	0.8	0.8	0.8	0.0	20.1	1.3
5	pluronic	0.8	0.8	0.8	0.0	20.6	1.0
6	agar&pluronic	0.8	0.8	0.8	0.0	21.7	1.2
7	water	0.8	0.8	0.8	0.0	22.1	0.8
8	fish_gelatin	0.8	0.8	0.8	0.0	24.5	1.6
9	agar	0.7	0.8	0.7	0.0	25.4	1.0

Table 1. Summary of quantitative distortion metrics across tested conditions. Mean scaling factors (S_x , S_y), anisotropy, contraction percentage, and RMS residuals are reported. Data confirm that neither hydrogel selection (e.g., agar, alginate, fish gelatin) nor surfactant supplementation (Pluronic, urea) reproducibly reduced meniscus curvature.

These results led us to conclude that additive-based strategies (either by selecting different hydrogel bases such as agar or alginate, or by supplementing with surfactants) could not overcome the problem. This realization ultimately motivated the development of HYDRA, which addresses meniscus distortion at its source.

References:

- Hancock, M. J., Yanagawa, F., Jang, Y. H., He, J., Kachouie, N. N., Kaji, H., & Khademhosseini, A. (2012). Designer hydrophilic regions regulate droplet shape for controlled surface patterning and 3D microgel synthesis. *Small (Weinheim an der Bergstrasse, Germany)*, 8(3), 393–403. <https://doi.org/10.1002/sml.201101745>
- Mihajlovic, M., Wyss, H. M., & Sijbesma, R. P. (2018). Effects of Surfactant and Urea on Dynamics and Viscoelastic Properties of Hydrophobically Assembled

Supramolecular Hydrogel. *Macromolecules*, 51(13), 4813–4820.
<https://doi.org/10.1021/acs.macromol.8b00892>

Modification 3

No modifications.

Reviewer 1

Comment 1.1

It is well established that cellular behavior like gene expression, drug responsiveness and morphology is strongly influenced by mechanical cues such as substrate stiffness. Therefore, the development of culture substrates that more accurately mimic the in vivo cell environment, while remaining compatible with standard high-throughput liquid handling and imaging systems in a well plate-based cell culture format, represents a promising strategy for enhancing the predictive power of in vitro drug screening assays. The novel approach presented here, including the quality control of the fabricated well plates, appears to be both feasible and practical. However, the proposed system is only suited for applications relying exclusively on imaging-based analysis. This limitation arises from the challenge of separating cells proliferating on the rigid surface adjacent to the hydrogel from the cells on top of the hydrogel when extracting RNA, proteins or retrieving cells for techniques like flow cytometry.

Response 1.1

We thank the reviewer for this observation. Indeed, the current implementation of HYDRA was conceived primarily as an imaging-compatible platform for high-throughput drug screening rather than as a universal culture system for all downstream analyses. As the reviewer points out, the residual rim of glass/plastic outside the hydrogel area complicates bulk extractions of RNA, proteins, or cells for flow cytometry. This is a limitation we explicitly acknowledged in the Discussion (lines 385–388).

We have now revised the Introduction and the Discussion to make this scope clear from the outset. HYDRA is designed to address the gap between biomimetic cell culture substrates and compatibility with automated, image-based high-content screening, which is increasingly used in biotech pipelines (e.g., cell painting, fluorescent biosensors, live-cell imaging). While the present version is optimized for these imaging-based applications, future iterations could incorporate strategies to extend hydrogel coverage across the entire well and thereby enable RNA/protein extraction.

Modification 1.1

- **Old text (Line 92-94):** Thus, HYDRA leverages established HTS automation and microscopy infrastructure, delivering physiologically relevant, easily implementable substrates to biotech screening programs without new capital investments.
 - **New text (Line 113-116):** ... new capital investments. *Importantly, HYDRA targets image-based drug screening assays, where the combination of biomimetic substrates and high-resolution microscopy can significantly enhance predictive power. While its current configuration is not designed for bulk RNA/protein extraction, it addresses the pressing need for imaging-compatible, high-throughput culture formats.*
 - **Old text (Line 344):** In this study, we asked, 'What would it look like if it were simple to have biomimetic elements in HTS?'
 - **New text (Line 389-390):** In this study, we asked, "What would it look like if it were simple to have biomimetic elements in HTS, *specifically tailored to the requirements of imaging-based drug screening assays?*"
-

Comment 1.2

Overall, the described methods are generally comprehensive and appropriately detailed, but it is recommended to also report the cell seeding densities and passage numbers used in the experiments.

Response 1.2

We have now included details on the cell passage number and the seeding density in the Materials and methods section, where they were previously missing.

Modification 1.2

- **Old text (Line 529-530):** Cultures were maintained below confluency and split at 70% confluence.
- **New text (Line 608-609):** Cultures were maintained below confluency and split at 70% confluence. *For all subsequent experiments, cells between passages 50 and 70 (indicated as P50-70) were used.*

- **Old text (Line 559-560):** Complementary to the holographic assay, a 96-well platform was used to capture static widefield images of cells on the gels at 24 and 48 h after drug treatment.
- **New text (Line 651-652):** Complementary to the holographic assay, a 96-well platform was used to capture static widefield images of cells (*P50-P70, 5×10^4 cells mL⁻¹*) on the gels at 24 and 48 h after drug treatment.

- **Old text (Line 612-613):** Cells on gels cast in 384-well plates (4681, Corning) were analyzed using long-term widefield acquisition and static confocal fluorescence imaging.
- **New text (Line 703-704):** Cells (*P50-P70, 1×10^3 cells mL⁻¹*) on gels cast in 384-well plates (4681, Corning) were analyzed using long-term widefield acquisition and static confocal fluorescence imaging.

Comment 1.3

The authors cite references in conjunction with statements that pertain to the results or investigations of the current study. It should be clearly distinguished which findings originate from the present work and which are based on previously published literature (Line 91, Line 131).

Response 1.3

In the original version, we intended to emphasize that our experimental results were consistent with values and trends reported in the literature. However, we recognize that the phrasing and use of references at those points (Lines 91 and 131) could give the impression that we were attributing our own results to prior studies.

To address this, we have removed the in-text references from those specific result descriptions. Instead, we now report only our experimental findings in the Results section and have shifted the discussion of consistency with published literature to the Discussion section, where it is

more appropriate. This change ensures that the distinction between our work and prior work is unambiguous, while still acknowledging the broader scientific context of our results.

Modification 1.3

- **Old text (Line 88-91):** We validated this approach using engineered fluorescent HaCaT epithelial cells cultured in both 96- and 384-well plates (Fig. 1c–e), demonstrating robust adherence, normal proliferation, and suitability for high-resolution, imaging-based drug-response assays [24].
 - **New text (Line 107-110):** ...demonstrating robust adherence, normal proliferation, and suitability for high-resolution, imaging-based drug-response assays.

 - **Old text (Line 130-131):** Independent of temperature and concentration, all FG solutions formed gels within 72 h (Fig. S1) [26].
 - **New text (Line 153-154):** Independent of temperature and concentration, all FG solutions formed gels within 72 h (Fig. S1).

 - **Old text (Line 363-365):** However, our method could be extended to other materials with liquid hydrogel precursors (also photocrosslinkable) to adjust the hydrogel stiffness and cell-binding properties, such as hyaluronic acid, alginate, or methacryloyl fish gelatin.
 - **New text (Line 419-422):** However, our method could be extended to other materials with liquid hydrogel precursors (also photocrosslinkable) to adjust the hydrogel stiffness and cell-binding properties, such as hyaluronic acid, alginate, or methacryloyl fish gelatin. *Importantly, the rheological trends we observed for fish gelatin, namely, the increase in viscosity with concentration and its fluidity at room temperature, are in good agreement with established reports on cold-water fish gelatin systems, which consistently describe lower gelation and melting points compared to mammalian counterparts [26, 27].*

 - **Old text (Line 383-384):** The extra layer at the bottom of the plate did not affect the standard imaging instruments, and the data quality was sufficient to automate data analysis in ways similar to current state-of-the-art cell painting assays and image analysis pipelines.
 - **New text (Line 441-444):** The extra layer at the bottom of the plate did not affect the standard imaging instruments, and the data quality was sufficient to automate data analysis in ways similar to current state-of-the-art cell painting assays and image analysis pipelines. *The adhesion and proliferation of engineered HaCaT cells that we observed on HYDRA gels are consistent with the well-established behavior of this cell line, which is known to maintain strong adhesion, proliferative capacity, and compatibility with imaging across a range of substrates [24].*
-

Comment 1.4

Abbreviations should be introduced before first use (for example Fig. 3: FOV, Line 113: TG).

Response 1.4

We have now introduced the abbreviations at their first occurrence in the text.

Modification 1.4

- **Old text (Line 112-113):** We cross-linked FG chemically using TG ...
 - **New text (Line 134-135):** We cross-linked FG chemically using *microbial transglutaminase* (TG) ...
-

Comment 1.5

(Line 152) The authors mention "hydrogels displayed physiologically relevant stiffness". What is regarded as physiological stiffness is highly dependent on the specific tissue/ cell type used. Please specify.

Response 1.5

The term "physiologically relevant stiffness" indeed varies depending on the tissue and cell type under consideration. For keratinocytes, native skin tissues range from the hypodermis ($E \sim 2$ kPa) to the dermis ($E \sim 35$ kPa) (Guimarães et al., 2020). HaCaT keratinocytes have been successfully cultured on hydrogels with $E \sim 3$ kPa, a stiffness shown to support both adhesion and traction generation (Schneider & Haugh, 2009). In addition, the normal dermis is often modeled around $E \sim 8$ kPa, while higher values are typically associated with fibrotic tissue (Laly et al., 2021). Based on this, the hydrogels used in our study ($E \sim 3$ kPa for 10% FG) lie at the lower end of the physiological stiffness range for skin and are consistent with conditions previously reported for HaCaT keratinocytes. To address this point more clearly, we have added a new section below to further clarify this aspect.

New references:

- Guimarães, C.F., Gasperini, L., Marques, A.P. *et al.* The stiffness of living tissues and its implications for tissue engineering. *Nat Rev Mater* 5, 351–370 (2020). <https://doi.org/10.1038/s41578-019-0169-1>.
- Schneider, I. C., Hays, C. K., & Waterman, C. M. (2009). Epidermal growth factor-induced contraction regulates paxillin phosphorylation to temporally separate traction generation from de-adhesion. *Molecular biology of the cell*, 20(13), 3155–3167. <https://doi.org/10.1091/mbc.e09-03-0219>.
- Ana C. Laly et al., The keratin network of intermediate filaments regulates keratinocyte rigidity sensing and nuclear mechanotransduction. *Adv.7,eabd6187(2021)*. DOI:10.1126/sciadv.abd6187.

Modification 1.5

- **Old text (Line 152-153):** We also found that the dispensed hydrogels displayed physiologically relevant stiffness when cross-linked with TG.
- **New text (Line 175-177):** We also found that hydrogels dispensed have reached stiffness ranges ($E \sim 1.5\text{--}6$ kPa) within the physiological range from single digit to tens

of kPa reported for skin tissues (Guimarães et al., 2020) and are consistent with conditions of stiffness already used for HaCaT keratinocytes (Schneider & Haugh, 2009; Laly et al., 2021).

Comment 1.6

(Line 223) “Three concentrations of nocodazole (12.5, 25, and 50 ng mL⁻¹, shown in Fig. 4a-b-c)”. Figure 4 a and b show holographic images of HaCaT cultured in 0.1% DMSO.

Response 1.6

We apologize for the confusion in referencing the figures. Indeed, Figure 4a–b displays the DMSO vehicle control, while Figure 4c shows HaCaT cells exposed to nocodazole (50 ng mL⁻¹). To avoid any ambiguity, we have revised the manuscript text to clearly indicate the role of the controls and the drug-treated condition, while maintaining the original figure order. The revised passage now reads:

Modification 1.6

- **Old text (Line 222-227):** We explored the range of drug concentrations known to affect the HaCaT cell cycle. Specifically, we chose three concentrations of nocodazole (12.5, 25, and 50 ng mL⁻¹, shown in Fig. 4a-b-c) and three concentrations of paclitaxel (0.5, 2.5, and 12.5 ng mL⁻¹ paclitaxel, as shown in Supplementary Fig. S6a-b-c). We included vehicle controls (0.1% DMSO in full culture media), where we expected no effect on the cell cycle, as well as positive controls, in which cells were exposed to a deadly concentration of an antibiotics to which HaCaT cells were not resistant (geneticin) (Supplementary Fig. S7a-b).
 - **New text (Line 257-263):** We explored the range of drug concentrations known to affect the HaCaT cell cycle. *Representative holographic images are shown in Figure 4 and Figure S6-S7: vehicle controls (0.1% DMSO at 0 h and 48 h, Fig. 4a–b, Supplementary Fig. S7), positive controls (Geneticin, Supplementary Figs. S6a-b, S7), nocodazole-treated cells (50 ng mL⁻¹, Fig. 4c), and paclitaxel-treated cells (Supplementary Fig. S6c). Specifically, we tested three concentrations of nocodazole (12.5, 25, and 50 ng mL⁻¹, Fig. 4c and Supplementary Fig. S8a) and three concentrations of paclitaxel (0.5, 2.5, and 12.5 ng mL⁻¹, Supplementary Figs. S6c, S8b).*
-

Comment 1.7

(Figure 3e) How is flatness of FG hydrogels displayed in that figure?

Response 1.7

In our analysis, flatness refers to the variation in thickness measurements taken at multiple positions across the surface of each hydrogel. For every sample, five field of views (FOVs) were imaged on the top surface, and the distribution of these values reflected the degree of flatness. We have now revised the caption of Figure 3 to clarify this definition.

Modification 1.7

- **Old text (Line 205-209):** (e) Assessment of thickness and flatness of FG hydrogels at three different concentrations (5% - orange squares, 10% - purple dots, and 20% w/v - black diamonds) in a hydrated state. The data corresponds to a single experiment (3 concentrations, n=5 samples for each concentration). For each sample, 5 FOVs were taken to measure the z-position and calculate thickness. Data are presented as mean \pm s.e.m.
 - **New text (Line 239-244):** (e) Assessment of thickness at three different concentrations of FG (5% - orange squares, 10% - purple dots, 20% w/v - black diamonds) in the hydrated state. The data correspond to a single experiment (3 concentrations, n=5 samples per concentration). For each sample, *five field of views (FOVs) were imaged on the top surface to measure the z-position and calculate thickness. The mean thickness per sample is reported, while flatness is reflected by the consistency of thickness values across the five fields of view.* Data are presented as mean \pm s.e.m.
-

Comment 1.8

(Line 325) What do the authors mean by “it remains viable for two months”?

Response 1.8

By “viable for two months”, we mean that hydrogels cast with HYDRA method can be stored in multi-well plates at 4°C for up to two months while retaining their functionality to support cell culture. After storage for 2 months, cells were seeded and imaged 24 h later. Hydrogels were considered viable if they supported cell adhesion, as confirmed by fluorescence imaging of actin-labeled cells (Supplementary Fig. S13a). No additional viability assays were performed beyond this visual inspection. We revised the text accordingly to clarify this point.

Modification 1.8

- **Old text (Line 322-324):** Together, our results show that HYDRA can be used in various well-plate formats, from low throughput to HTS, and substrates, from glass to plastic, where it remains viable for two months (Supplementary Fig. S13).
 - **New text (Line 368-369):** Together, our results show that HYDRA can be used in various well-plate formats, from low throughput to HTS, and substrates, from glass to plastic, *and that hydrogel cast with this method can be stored at 4°C for up to two months while retaining their ability to support cell adhesion* (Supplementary Fig. S13).
-

Comment 1.9

More references should be included to support general statements throughout the manuscript. Where possible, older citations should be replaced or complemented with recent literature (for example: Line 55, Line 195, Line 75-80).

Response 1.9

We have revised the manuscript to ensure that appropriate references well support each general statement. Where possible, we have complemented classic citations with recent studies that confirm or extend these principles in contexts more directly relevant to our work.

Modification 1.9

- **Old text (Line 53-55):** ... early automated high-throughput screening (HTS) - have predominantly relied on rigid substrates such as plastic and glass, which distort tissue architecture, alter mechanical and metabolic signaling, and impair interactions between cells and the extracellular matrix (ECM), compromising assay relevance ([7] → Kapałczyńska et al., 2018).
- **New text (Line 54-56):** ..., alter mechanotransduction and cellular metabolism, and bias cell–ECM interactions, thereby compromising physiological assay relevance relative to compliant 2D hydrogel beds (Kapałczyńska et al., 2018; Yeung, T. et al., 2005; Elosegui-Artola et al., 2016; Na Jing et al., 2018).

- **Old text (Line 194-195):** This result agrees with the literature, which sets the cut-off point for how deeply cells sense substrates at 10-20 μm ([22] → Buxboim et al., 2010).
- **New text (Line 228-229):** ... how deeply cells sense substrates at 10-20 μm (Buxboim et al., 2010, Tusan et al., 2018, and Hernandez-Miranda et al., 2024).

- **Old text (Line 75-80):** Recent efforts sought to overcome these limitations. Brooks et al. formed PEG-based hydrogels in 96-well plates through photocrosslinking, yielding tunable stiffness, yet the thick (500–1000 μm) gels compromised optical resolution. At the other end of the spectrum, Machillot et al. utilized a layer-by-layer coating to create single micron hydrogel layers, but these coatings were insufficiently thick to shield cells from the underlying stiff plastic substrate, influencing cellular responses.
- **New text (Line 82-94):** Achieving uniform thin hydrogel layers within multiwell HTS plates remains technically challenging: the meniscus at well edges creates curvature that disrupts cell attachment, homogeneous growth, and high-resolution microscopy ([19] → Trask, 2018; Auld et al., 2020; Nienhaus et al., 2023). PEG and hyaluronic acid-based hydrogels have been fabricated in 96-well plates through photocrosslinking, yielding tunable stiffness and viscoelasticity, yet the thick (500–1000 μm) gels compromised optical resolution ([20] → Brooks et al., 2018, Skelton et al., 2024). Machillot et al. established an automated LbL process in microplates yielding optically transparent PEM films with thicknesses from a few tens of nanometers to a few micrometers, aided by plate tilting and ~10% over-aspiration to improve uniformity ([21] → Machillot et al., 2018). However, effective mechanical shielding of adherent cells from the rigid polystyrene generally requires soft layers above mechanosensing length-scales, on the order of ~10–20 μm for single cells and considerably thicker for colonies (half-maximal depth response ~54 μm at 1 kPa) [22 → Buxboim, et al., 2010, Janmey et al., 2020]. Scaling LbL to such thicknesses in 96/384-well formats would entail impractically many deposition/rinse cycles with

elevated risk of defects and delamination; moreover, increased chemical crosslinking (e.g., EDC/NHS) stiffens PEMs and counteracts the desired compliant interface [21].

New references:

1. Yeung, T. et al., “Effects of substrate stiffness on cell morphology, cytoskeletal structure, and adhesion”. *Cell motility and the cytoskeleton*, 60(1), 24–34 (2005). DOI: <https://doi.org/10.1002/cm.20041>
2. Elosegui-Artola et al., “Mechanical regulation of a molecular clutch defines force transmission and transduction in response to matrix rigidity”. *Nature cell biology*, 18(5), 540–548 (2016). DOI: <https://doi.org/10.1038/ncb3336>
3. Na, J., et al., “Extracellular matrix stiffness as an energy metabolism regulator drives osteogenic differentiation in mesenchymal stem cells”. *Bioactive materials*, 35, 549–563 (2024). DOI: <https://doi.org/10.1016/j.bioactmat.2024.02.003>
4. Tusan CG. et al., “Collective Cell Behavior in Mechanosensing of Substrate Thickness”. *Biophys J* 114, 1456–1468 (2018). DOI: <https://doi.org/10.1016/j.bpj.2018.03.037>
5. Hernandez-Miranda ML. *et al.* “Geometric constraint of mechanosensing by modification of hydrogel thickness and shape.” *J R Soc Interface* 21, 20230506 (2024). DOI: [10.1098/rsif.2024.0485](https://doi.org/10.1098/rsif.2024.0485)
6. M. L. Skelton et al., “Modular Multiwell Viscoelastic Hydrogel Platform for Two- and Three-Dimensional Cell Culture Applications”. *ACS Biomaterials Science & Engineering* 10(5), 3280-3292 (2024). DOI: <https://pubs.acs.org/doi/10.1021/acsbiomaterials.4c00312>
7. Janmey, P. A., Fletcher, D. A., & Reinhart-King, C. A. (2020). Stiffness Sensing by Cells. *Physiological reviews*, 100(2), 695–724. <https://doi.org/10.1152/physrev.00013.2019>

Comment 1.10

(Line 136) Please correct: “...storage modulus (G') and loss modulus (G'') ...”

Response 1.10

We have corrected the text accordingly.

Modification 1.10

- **Old text (Line 136):** ...storage modulus (G') and loss modulus (G'') ...
 - **New text (Line 159):** ...the storage modulus (G') and loss modulus (G'') ...
-

Comment 1.11

Have the authors analyzed the swelling behavior of the hydrogel in contact with culture medium over the full duration of the cell cultivation experiments? Additionally, have the authors investigated how far in advance the well plates can be prepared prior to cell seeding? In particular, how was the reported maximum storage duration of two months (Line 501) determined?

Response 1.11

We apologize for any confusion arising from our original wording that did not distinguish sufficiently from our cell culture and storage experiments.

All cell culture experiments reported in this work were performed using freshly prepared hydrogels, which were cast within 48 hours before use. The longest cultivation experiment lasted 48 h, and no hydrogel degradation or detachment was observed during this period. Thus, we initially did not consider the swelling behavior of the gels. To complement these experiments, we have now added profilometry data on dry gels (Supplementary Fig. S5c-d), in addition to the hydrated state initially reported. This enabled us to calculate the swelling ratio ($SR = h_{wet} / h_{dry}$). The results show a clear dependence on FG content, with $SR \approx 6.7$ for 5% FG, ≈ 4.1 for 10% FG, and ≈ 1.1 for 20% FG. These findings confirm that higher FG concentrations reduce swelling capacity, and that the larger hydrated thickness observed at 20% FG mainly reflects the higher solid content rather than extensive swelling.

Regarding storage, the reported “two months” does not refer to continuous culture but to the possibility of keeping crosslinked gels hydrated in PBS at 4 °C before use. After storage for up to two months under these conditions, the PBS was replaced with culture medium, and cells were seeded. Cells adhered and spread on these stored gels (Supplementary Fig. S13a), indicating that they retained their functionality after long-term storage. We have revised the manuscript to clarify this distinction between short-term swelling during culture and long-term storage in PBS before cell seeding.

Supplementary Fig. S5. Assessment of thickness, flatness, and swelling of fish gelatin (FG) hydrogels. ... (c) Representative 3D optical profilometry scan of a dry FG hydrogel showing surface topography and flatness. (d) Swelling ratios (h_{wet}/h_{dry}) calculated from hydrated and dry thickness measurements. Higher FG concentrations reduce swelling capacity, with ratios decreasing from ~ 6.7 (5% FG) to ~ 4.1 (10% FG) and ~ 1.1 (20% FG).

Modification 1.11

- **Old text (Line 501):**
After crosslinking, the plates were rehydrated in PBS, sealed with parafilm (PM-992, Bemis), and stored at 4 °C for up to two months.
 - **New text (Line 578-579):**
... up to two months. *Before use, PBS was replaced with cell culture medium and cells were seeded, confirming that the gels retained their ability to support cell adhesion after storage.*

 - **Old text (Line 189-194):** Next, to verify compatibility with standard microscopy, we characterized hydrogel thickness and flatness by embedding fluorescent beads within the gels and seeding epithelial cells expressing a red fluorescent actin marker (LifeAct™) (Fig. 3d). Using a 20× air objective (NA 0.75), we captured two images, one at the top of the gel and the other at the bottom of the well, by focusing on cell actin and the fluorescent beads. The samples exhibited an expected dose-dependent increase in thickness, from ~10 μm to ~30 μm, and inter-group reproducibility (Fig. 3e and Supplementary Fig. S5).
 - **New text (Line 224-228):** ... and inter-group reproducibility (Fig. 3e and Supplementary Fig. S5). *To assess whether this effect was linked to gel swelling, we also measured dry gel thickness by optical profilometry (Supplementary Fig. S5c-d). The resulting swelling ratios decreased with increasing FG concentration (~6.7 at 5%, ~4.1 at 10%, and ~1.1 at 20% FG), indicating that the greater hydrated thickness of 20% gels mainly reflects their higher solid content rather than swelling.*

 - **New text (Line 623-635 – new method section):** **“Morphological characterization of robot-cast gels in dried states”.** *The radius, thickness, and flatness of the fabricated hydrogels with cylindrical shapes were characterized in dried forms by optical profilometry. Hydrogel samples featuring 5%, 10%, and 20% w/v fish gelatin were cast on glass coverslips (VD12260Y1A.01, Knittel Glasbearbeitungs GmbH) using the scalable protocol previously described and left to crosslink overnight at 37 °C. The samples were then let dry for at least one day at room temperature (24 °C) and atmosphere. The samples in the dried state were then imaged with a Wyko NT9300 white light profilometer (Veeco) using a 20X 0.4 NA CF IC Epi Plan DI Mirau Interferometry Objective (Nikon) in vertical scanning interferometer (VSI) mode. The data was processed, analyzed, and 3D rendered using Vision 4.2 software (Veeco). At least three hydrogels per condition were imaged, and x- and y-cuts were taken in correspondence with the center of each gel to estimate thickness and flatness. Spherical aberration in the optical setup and substrate curvature were corrected using a second-order polynomial normalization using the thickness values of the glass substrate around the hydrogels.*
-

Reviewer 2

Comment 2.1

Introduction. The manuscript presents HYDRA, a novel method for fabricating planar hydrogels in multiwell plates using robotic liquid handling, with the aim of enhancing the physiological relevance of high-throughput screening (HTS). The authors make a case for the need to improve substrate biomimicry in early drug testing pipelines, highlighting limitations in current glass/plastic systems and the impracticality of more complex 3D models like organoids and organs-on-chip at HTS scale. The motivation is real, though the introduction could better acknowledge concurrent innovations in microfabricated 2.5D systems, which share overlapping goals. The rationale for choosing fish gelatin over more established materials such as PEG or collagen could also benefit from deeper comparative justification early on.

Response 2.1

We thank the reviewer for this and the following suggestions. We agree that microfabricated 2.5D systems are a critical parallel development in efforts to enhance biomimicry within scalable screening formats. In the revised manuscript, we have expanded the Introduction to better situate HYDRA within this broader landscape. Specifically, we now acknowledge seminal work integrating electrical and mechanical readouts, as well as approaches that leverage microfabricated substrates to systematically tune stiffness and geometry in high-throughput-compatible arrays.

Modification 2.1

- **Old text (Line 65-69):** ...Organs-on-chip similarly capture complex tissue functionality through precise control of mechanical and biochemical conditions within microfluidic platforms. Their planar configuration facilitates high-resolution microscopy and optical clarity. Yet, the intricate and custom nature of microfabrication currently prevents scalable production, severely restricting their practical implementation within automated, high-throughput pipelines...
- **New text (Line 70-78):** ...within automated, high-throughput pipelines. *In parallel with 3D organoids and microfluidic organs-on-chips, microfabricated “2.5D” platforms have matured to engineer laminar, quasi-three-dimensional tissues on planar substrates that preserve optical access and enable integrated functional readouts. Notable examples include multimaterial-printed cardiac devices with embedded strain sensors for continuous mechanical measurements (Lind et al., 2017), high-throughput electrode arrays enabling electrical phenotyping across many conditions (McConnell et al., 2012; Feyen et al., 2020), and microarray platforms with tunable substrate stiffness for parallel mechanobiology assays (Kobel et al., 2012). Collectively, these 2.5D strategies improve control of geometry, mechanics, and electrophysiology, but often rely on bespoke microfabrication workflows or specialized materials that complicate scale-up in standard HTS pipelines.*

New references:

- Lind, J. U., et al., “Instrumented cardiac microphysiological devices via multimaterial three-dimensional printing”. *Nature materials*, 16(3), 303–308, (2017). DOI: <https://doi.org/10.1038/nmat4782>
 - McConnell, E. R., et al. “Evaluation of multi-well microelectrode arrays for neurotoxicity screening using a library of compounds with known effects on neuronal activity.” *Neurotoxicology* 33.5 (2012): 1048–1057. <https://doi.org/10.1016/j.neuro.2012.05.001>
 - Feyen, D. A. M., et al. “Metabolic maturation media improve physiological function of human iPSC-derived cardiomyocytes.” *Cell Reports* 32.3 (2020): 107925. <https://doi.org/10.1016/j.celrep.2020.107925>
 - Kobel, S. & Lutolf, M. Fabrication of PEG Hydrogel Microwell Arrays for High-Throughput Single Stem Cell Culture and Analysis. *Methods in molecular biology* (Clifton, N.J.) 811, 101-12 (2012). https://doi.org/10.1007/978-1-61779-388-2_7
-

Comment 2.2

Results. The authors validate HYDRA using rheological assays, finite element modeling, and imaging-based drug testing. The robustness of their mechanical characterization is commendable, particularly their quantitative handling of viscosity and stiffness, and their demonstration of control over hydrogel flatness via dispensing parameters. The biological data, including dose-response assays with nocodazole and paclitaxel, are convincing and confirm that the thin gels support normal cell growth and drug responsiveness. However, the differences in IC₅₀ values between plastic and hydrogel substrates are relatively modest, which raises the question of how much functional advantage HYDRA offers over standard coatings. The authors report increased final confluency on hydrogels, but whether this improvement translates to better predictive power in real-world drug screens remains speculative and unquantified.

Response 2.2

We failed to clearly explain in the original manuscript our goal for these experiments. Our primary goal in designing this study was to demonstrate a proof-of-concept for imaging-based drug assays on thin, mechanically defined hydrogels. We selected HaCaT keratinocytes specifically because they stably express the cell-cycle reporters required for our quantitative analyses, not to demonstrate any specific pharmacological property of these cells.

At the same time, the modest IC₅₀ differences we observe between plastic (1.6 nM) and hydrogels (1.8 nM) are in line with previously reported variations. For example, HeLa cells treated with paclitaxel exhibit values of 2.3 nM (Zustiak et al., 2014) on 1 kPa gels versus 1.9 nM on glass (Peng et al., 2014), a difference of 0.4 nM, comparable to ours. In other cell types, however, stiffness effects can be larger: MCF-7 cells shift from 3.2 nM at 1 kPa to 8.3 nM at 100 kPa, while PC3 cells vary only minimally (13.0 vs. 12.9 nM across the same range) (Zustiak et al., 2014). These examples emphasize that the impact of substrate mechanics is highly cell-type dependent.

Taken together, our results show that HYDRA provides a robust, cell-compatible platform that maintains drug responsiveness in HaCaT cells without introducing artificial shifts. At the same

time, the increased confluency on hydrogels highlights the potential for improved reproducibility in long-term assays. We agree with the reviewer that large-scale screens across diverse cell types will be essential to fully establish the predictive power of HYDRA, and we now stress this in the revised Discussion.

Modification 2.2

- **Old text (Line 383-384):** The extra layer at the bottom of the plate did not affect the standard imaging instruments, and the data quality was sufficient to automate data analysis in ways similar to current state-of-the-art cell painting assays and image analysis pipelines. The adhesion and proliferation of engineered HaCaT cells that we observed on HYDRA gels are consistent with the well-established behavior of this cell line, which is known to maintain strong adhesion, proliferative capacity, and compatibility with imaging across a range of substrates [24].
- **New text (Line 444-449):** ... imaging across a range of substrates [24]. *In this proof-of-concept study with HaCaT keratinocytes, chosen for their stable cell-cycle reporters, the IC₅₀ differences between plastic and hydrogel substrates were modest (1.6 vs. 1.8 nM). Similar small shifts have been reported in HeLa cells (2.3 nM at 1 kPa vs. 1.9 nM on glass; Peng et al., 2014), while other lines show larger or negligible effects depending on stiffness (Zustiak et al., 2014). These findings indicate that HYDRA preserves drug responsiveness while offering a physiologically relevant and reproducible culture substrate.*

New references:

- Peng, X., Gong, F., Chen, Y. *et al.* Autophagy promotes paclitaxel resistance of cervical cancer cells: involvement of Warburg effect activated hypoxia-induced factor 1- α -mediated signaling. *Cell Death Dis* 5, e1367 (2014). <https://doi.org/10.1038/cddis.2014.297>
 - Zustiak, S., Nossal, R., & Sackett, D. L. (2014). Multiwell stiffness assay for the study of cell responsiveness to cytotoxic drugs. *Biotechnology and bioengineering*, 111(2), 396–403. <https://doi.org/10.1002/bit.25097>
-

Comment 2.3

Discussion. The discussion is thoughtful and transparent, particularly in acknowledging the edge effects from residual plastic rims and potential limitations in biochemical extractions. The authors are right to emphasize the compatibility of HYDRA with standard HTS infrastructure, which is indeed one of its most attractive features. However, the central claim that HYDRA offers a better balance between biomimicry and scalability needs to be better substantiated. For instance, while flatness is shown to be improved, how this affects downstream image analysis quality or assay reproducibility is not explicitly tested. Similarly, while the gel thickness is in a biologically meaningful range, it's unclear how uniform this remains across large batches or over time? The manuscript would benefit from more quantitative comparison to other thin hydrogel methods, particularly in terms of throughput, cell behavior variability, and imaging performance.

Response 2.3

We agree that our central claim, that HYDRA strikes a better balance between biomimicry and scalability, requires further substantiation, and we have revised the manuscript to address i) imaging artifacts in micro-wells in which a hydrogel meniscus formed; ii) the effect of environmental variables; and, iii) a comparison with existing platforms across throughput levels.

We now include a supplementary figure (Supplementary Fig. S4d-e) showing a full z-stack of a meniscus-containing well seeded with cells. This illustrates two critical points: (i) cells plated on meniscus substrates are not all in the same focal plane, which complicates downstream image analysis and segmentation, and (ii) the cell distribution is highly non-uniform, with cells concentrating at the edges of the well. By contrast, HYDRA's flat films eliminate these artifacts, ensuring that cells remain in a single focal plane and uniformly distributed.

Fig. S4. Modelling of the HYDRA dispensing method. [...] (d) Representative z-stack reconstruction of a hydrogel with full meniscus inside a high-throughput well, displayed in orthogonal views (xy, xz, yz). The hydrogel containing fluorescent beads is shown in magenta, while cells (actin) appear in gray. (e) Zoom-in of the xy plane from panel d, highlighting the curved meniscus across the entire well area.

We have expanded the Discussion to emphasize how environmental variables (temperature, humidity, and evaporation at low casting volumes) can influence hydrogel uniformity. In our hands, HYDRA films exhibit consistent thickness across plates when fabricated under stable environmental conditions; however, we acknowledge that more systematic quantification across large production batches and over extended storage times would be valuable. We therefore frame this as a key area for future industry–academic collaboration, particularly in contexts where environmental control systems differ academia and industry.

Furthermore, we have revised the Discussion to explicitly compare HYDRA with other low-, medium-, and high-throughput hydrogel-based approaches. This comparison covers (i) physical parameters such as stiffness and thickness, (ii) scalability in terms of throughput and automation compatibility, and (iii) biological readouts including cell distribution, viability, and variability in behavior. This framing situates HYDRA within a broader landscape, highlighting where it offers unique advantages (flatness, reproducibility, and HTS integration) while acknowledging areas where further benchmarking is warranted.

Modification 2.3

- **Old text (Line 168-188):** To optimize hydrogel fabrication by minimizing meniscus formation and maximizing substrate coverage, we used finite-element modeling (FEM) in COMSOL. [...] Therefore, we selected 12 μL for the experiments in this study, which reduced coverage by $\sim 20\%$ and increased flatness by 4.5 times concerning the meniscus-inducing conditions (Fig. 3 b and Supplementary Fig. S4). Accordingly, we automated FG hydrogel fabrication in 96-well plates using an Integra liquid-handling robot, dispensing and immediately re-aspirating 12 μL of precursor solution (visualized using red colorant) to leave only a thin boundary layer. This procedure required about 10 minutes, well within the 40-minute gelation threshold established earlier (Fig. 3c; Supplementary Movies S3 and S4).
- **New text (Line 213-218):** ... established earlier (Fig. 3c; Supplementary Movies S3 and S4). *To experimentally visualize the meniscus of the liquid, we imaged full meniscus geometries in z-stacks across high-throughput wells (Supplementary Fig. S4d–e). These reconstructions, shown (d) in orthogonal planes (xy, xz, yz) and (e) at higher magnification in the xy view, highlight the extent of curvature and its impact on planar coverage. Cellular actin is displayed in gray and fluorescent beads–embedded hydrogels in magenta.*

Modifications related to the Discussion are detailed in the Response and Modifications 2.4 below, since they shared a common thread.

Comment 2.4

Materials & Methods. The methods section is ok, with protocols and a range of technologies spanning computational modeling, rheology, and automated microscopy. The use of off-the-shelf robotics is a strength, but it's unclear how easily these workflows could be implemented in other labs without detailed calibration. While the authors stress the minimal infrastructural burden, liquid handling at the microliter scale with precise aspiration timing remains non-trivial. Additional benchmarking across different robotic platforms would have strengthened claims of generalizability.

Response 2.4

We agree that liquid handling at the microliter scale requires careful optimization, and we have revised both the Materials and Methods and Discussion to clarify our approach and scope.

First, in this project we deliberately began with an accessible, entry-level robotic platform (OT-2, Opentrons), which can be readily implemented in many academic labs and supports formats up to 96-well plates. This allowed us to demonstrate that HYDRA can be executed using open-source, low-cost systems. To address higher-throughput formats such as 384-well plates, we subsequently implemented HYDRA on an industry-standard system (Assist Plus, INTEGRA), which is routinely employed in biotech settings for precise liquid handling at microliter scales. This stepwise approach, from accessible robotics to pharma-grade automation, illustrates both the entry point for academic users and the scalability required for industrial pipelines.

In the Materials and Methods section, we now explicitly state at the beginning of each subsection which robotic platform was used and for which specific purpose (e.g., OT-2 for proof-of-concept in low-throughput formats, INTEGRA Assist Plus for 96- and 384-well

scalability). In the *Discussion*, we have clarified that while HYDRA can be implemented with entry-level systems. Even the industrial platform we had access to reach its limits with 384 well plates, as we transparently acknowledge in the original manuscript. At the same time, robots with increased precision must be operated in environment with tighter controls of environmental variables and stringent QC on raw materials that we can achieve in academia. We concluded that benchmarking across diverse robotic platforms would be a promising future direction that would benefit from close industry–academic collaboration.

Modification 2.4

- **Old text (Line 346-350):** Our answer is HYDRA, which is a scalable and automated approach for fabricating thin hydrogel films at the bottom of commercial HTS plates. To do so, throughout this study, we sought material composition and fabrication strategies that would be simple to implement not only in academic research frameworks but also in industry-level scenarios.
- **New text (Line 395-404):** ... but also in industry-level scenarios. *In this study, we illustrate this dual accessibility by first implementing HYDRA on an entry-level OT-2 platform (Opentrons), which demonstrates feasibility in academic labs with minimal infrastructural burden. To address higher-throughput formats, we then employed the INTEGRA Assist Plus, a platform widely adopted in biotech settings. This progression highlights how HYDRA can bridge academic proof-of-concept studies and industry-scale screening pipelines. Similarly, environmental variables, such as temperature and humidity, can affect gel casting (Mansoury M. et al. 2021, Xie, et al. 2019, Feng M. et al. 2024), especially at the low volumes used for HYDRA fabrication. A systematic comparison of liquid handling automation and environmental control systems, ranging from other open-source to industry-grade, represents an important future direction which will need industry collaboration.*
- **Old text (Line 472-475):** An open-source pipetting system (OT-2 robot, Opentrons) was used to fabricate micrometer-thick planar hydrogels. Specifically, a liquid handling robot pipeline was implemented using the Opentrons Protocol Designer (Version 7.0.0, Opentrons) to cast and re-aspirate the hydrogel precursor solutions, leaving planar cylindrical hydrogels on the plastic substrates.
- **New text (543-546):** An open-source, *entry-level* pipetting system (OT-2 robot, Opentrons) was used to demonstrate proof-of-concept fabrication of micrometer-thick planar hydrogels in formats up to 96-well plates. This platform was deliberately chosen to illustrate HYDRA's accessibility for academic laboratories with minimal infrastructural burden.
- **Old text (Line 491-492):** FG hydrogels were fabricated in 96-well plates using an Assist Plus robot with a 3-position deck (4520, Integra Biosciences) to enable scalable hydrogel production.
- **New text (Line 565-569):** *To enable scalable hydrogel production in high-throughput formats, a commercial robotic system (Assist Plus, INTEGRA Biosciences) was employed. This platform, designed for high-throughput applications, including biotech settings, was applied to fabricate hydrogels in 96-well and 384-well plates with microliter-scale precision. FG hydrogels were fabricated ...*

New references:

- Mansoury M et al., “The edge effect: A global problem. The trouble with culturing cells in 96-well plates”. *Biochem Biophys Rep.*, 2021. DOI: 10.1016/j.bbrep.2021.100987.
 - Xie, et al., “Protocols of 3D Bioprinting of Gelatin Methacryloyl Hydrogel Based Bioinks”. *J. Vis. Exp.*, (2019), DOI:10.3791/60545.
 - Mingwei Feng, “Coupling effect of curing temperature and relative humidity on the unconfined compressive strength of xanthan gum-treated sand”, *Construction and Building Materials*, 2024, DOI: <https://doi.org/10.1016/j.conbuildmat.2024.138224>.
-

Comment 2.5

In conclusion, this is a good contribution. The work is technically acceptable and biologically relevant, though some claims regarding its practical impact in HTS could be more rigorously supported.

Response 2.5

We agree that clarifying the scope of HYDRA’s practical impact within high-throughput screening (HTS) is important. Our intent was not to claim universal applicability across all HTS modalities, but rather to highlight its specific value for imaging-driven high-content approaches, where the integration of biomimetic substrates with automated microscopy can substantially increase the physiological relevance of readouts. Furthermore, by selecting central fields of view, imaging-based HTS is inherently robust to HYDRA’s main limit: the remaining plastic rim.

In its present form, HYDRA is designed to address the gap between biomimetic cell culture substrates and compatibility with automated, image-based high-content screening, which is increasingly used in biotech pipelines (e.g., cell painting, fluorescent biosensors, live-cell imaging). While the present version is optimized for these imaging-based applications, future iterations could incorporate strategies to extend hydrogel coverage across the entire well and thereby enable RNA/protein extraction.

We have revised the Introduction and the Discussion to temper broad claims and to explicitly frame HYDRA as a platform optimized for image-based HTS applications, which are increasingly central in drug discovery pipelines (e.g., cell painting, phenotypic profiling, live-cell imaging).

Modification 2.5

- **Old text (Line 92-94):** Thus, HYDRA leverages established HTS automation and microscopy infrastructure, delivering physiologically relevant, easily implementable substrates to biotech screening programs without new capital investments.
- **New text (Line 113-116):** ... new capital investments. *Importantly, HYDRA targets image-based drug screening assays, where the combination of biomimetic substrates and high-resolution microscopy can significantly enhance predictive power. While its current configuration is not designed for bulk RNA/protein extraction, it addresses the pressing need for imaging-compatible, high-throughput culture formats.*

- **Old text (Line 344):** In this study, we asked, 'What would it look like if it were simple to have biomimetic elements in HTS?'
 - **New text (Line 389-390):** In this study, we asked, 'What would it look like if it were simple to have biomimetic elements in HTS, *specifically tailored to the requirements of imaging-based drug screening assays?*'
-

Reviewer 3

Comment 3.1

In this paper, Torchia et al present a novel method for consistent fabrication of hydrogels in high-throughput screening (HTS) plates using automated liquid handling. Using a combination of high-resolution imaging and multiple pharmacological assays performed on both 96 and 384 well plates, they also demonstrate the scalability of their technique and its potential applicability to in-vitro drug testing. Overall, the study is robust and well-planned; however, there are some aspects that need additional clarification/revision as listed below: Page 7 line 148- There seems to be a confusion between storage modulus and shear modulus, as the same notation G' is used for both here. Please clarify how the shear modulus was calculated for your analysis in fig 2 e&f, because it is different from storage modulus.

Response 3.1

We thank the reviewer for bringing this point to our attention. Indeed, we interchangeably used the terms storage modulus and shear modulus. We inadvertently did this, as in our materials $G' \gg G''$ meaning the materials are predominantly elastic, and the time sweep experiments were performed within LVR and at low frequency. Under these conditions, the storage modulus is indeed approximated to the shear modulus. However, to avoid any confusion, we have now adjusted Fig. 2d and f by indicating storage modulus on the y-axis, as the values were taken from G' derived from time sweep measurements.

Modification 3.1

Figure 2e and 2f have been modified as shown below:

Old text (162-166): (e) Shear modulus G' vs. FG concentration (5% - orange squares, 10% - purple dots, and 20% w/v - black diamonds). FG solutions were crosslinked with 2% w/v TG. Data are displayed as mean \pm s.e.m (n=3 samples). (f) Shear modulus G' vs. TG concentration (0.5% - orange squares, 1% - purple dots, and 2% w/v - black diamonds). 10% w/v fixed FG was used. Data are displayed as mean \pm s.e.m (n=3 samples).

New text (Line 186-190): (e) Storage modulus ... (f) Storage modulus ...

Comment 3.2

The authors have performed time and amplitude sweep as well as the variation of viscosity with shear rates. Was there a specific reason to not perform frequency sweep analysis?

Response 3.2

We thank the reviewer for the comment. Those tests were not performed because our primary goal in characterizing these hydrogels was to assess their stiffness and gelation kinetics, which were obtained through time sweep experiments. Since our crosslinked gels are mainly elastic (from time sweep experiment $G' \gg G''$ by several orders of magnitude), we anticipated weak and minimal moduli dependency on the frequency, and this information was not crucial in the context of characterizing the materials mechanics towards robotic handling or material stability, as opposed to amplitude sweep and flow sweep tests.

Supplementary Fig. S3e. Frequency sweep of 10% (w/v) fish gelatin hydrogel crosslinked with 2% (w/v) microbial transglutaminase (mTG). Storage modulus (G' , magenta) and loss modulus (G'' , gray) are shown as a function of frequency (1–100 Hz). The data demonstrate a frequency-independent elastic response with $G' \gg G''$, indicating the predominantly solid-like behavior of the hydrogel.

Modification 3.2

No modifications.

Comment 3.3

(Page 9 - Section 3.2) *The simulation of meniscus formation using FE modeling provides useful insights on how to identify the optimal dispensing volume. However, it may be useful to study the effect of extrusion speeds/flow rates on meniscus formation as well as the material viscosity is shear rate dependent.*

Response 3.3

In this simulation study, we assumed, based on rheological measurements, that the hydrogel precursors behaved as sufficiently homogeneous fluids under the dispensing conditions employed. For this reason, we did not explicitly account for the influence of extrusion speed or shear rate–dependent viscosity on meniscus formation. We fully agree, however, that incorporating these parameters would provide a more comprehensive description, particularly for complex or strongly non-Newtonian hydrogel systems. To acknowledge this point, we have added a sentence in the Discussion noting that future extensions of the model could incorporate flow-dependent physics, as explored in recent extrusion-based bioprinting frameworks (e.g., Chirianni, Vairo & Marino, *Meccanica*, 2024).

Modification 3.3

- **Old text (Line 389-393):** Compared to the layer-by-layer and photo-crosslinking approaches for obtaining hydrogels in HTS plates, we traded good coverage in favor of physiologically relevant thickness and high-res imaging compatibility. Because the final plate remains optically and chemically addressable, combining these and other approaches to generate full-well, more complex microenvironments in future versions of HYDRA will be possible.
- **New text (Line 458-461):** ... more complex microenvironments in future versions of HYDRA will be possible. *Furthermore, future refinements of the finite-element framework could incorporate flow-dependent physics, such as extrusion speed and shear-rate–dependent viscosity, similar to what has been recently explored in extrusion-based bioprinting models (e.g., Chirianni, Vairo & Marino, Meccanica, 2024), thereby extending the predictive power of this HYDRA computational model beyond volume-driven meniscus dynamics.*

New reference:

- Chirianni, F., Vairo, G. & Marino, M. Influence of extruder geometry and bio-ink type in extrusion-based bioprinting via an in-silico design tool. *Meccanica* 59, 1285–1299 (2024). <https://doi.org/10.1007/s11012-024-01862-7>
-

Comment 3.4

(Page 10, Fig 3e) The gel thickness is higher for increased concentrations of FG. Considering all gels have similar volumes, is this related to gel swelling properties? Some clarifications here may be useful.

Response 3.4

In the original submission, we had reported gel thickness only in the hydrated state, which may have led to ambiguity regarding the contribution of swelling. To clarify this point, we performed additional optical profilometry measurements on the dry gels (new Figs. S5C–D) and calculated the swelling ratio ($SR = h_{wet}/h_{dry}$).

The results show a clear trend: the swelling ratio decreases with increasing FG concentration, from ~6.7 at 5% FG to ~4.1 at 10% FG, and down to ~1.1 at 20% FG. These data indicate that gels with lower FG content undergo greater volumetric expansion upon hydration, whereas highly concentrated gels retain nearly the same thickness in wet and dry states. Thus, the larger thickness observed in hydrated 20% FG gels is not due to swelling but rather reflects their intrinsically higher solid content.

We have included these new data and clarified the explanation in the revised manuscript (Supplementary Figs. S5c–d).

Supplementary Fig. S5. Assessment of thickness, flatness, and swelling of fish gelatin (FG) hydrogels. ... (c) Representative 3D optical profilometry scan of a dry FG hydrogel showing surface topography and flatness. (d) Swelling ratios (h_{wet}/h_{dry}) calculated from hydrated and dry thickness measurements. Higher FG concentrations reduce swelling capacity, with ratios decreasing from ~6.7 (5% FG) to ~4.1 (10% FG) and ~1.1 (20% FG).

Modification 3.4

- **Old text (Line 189-194):** Next, to verify compatibility with standard microscopy, we characterized hydrogel thickness and flatness by embedding fluorescent beads within the gels and seeding epithelial cells expressing a red fluorescent actin marker (LifeAct™) (Fig. 3d). Using a 20× air objective (NA 0.75), we captured two images, one at the top of the gel and the other at the bottom of the well, by focusing on cell actin and the fluorescent beads. The samples exhibited an expected dose-dependent increase in thickness, from ~10 μm to ~30 μm, and inter-group reproducibility (Fig. 3e and Supplementary Fig. S5).
- **New text (Line 224-228):** ... and inter-group reproducibility (Fig. 3e and Supplementary Fig. S5). To assess whether this effect was linked to gel swelling, we

also measured dry gel thickness by optical profilometry (Supplementary Fig. S5c-d). The resulting swelling ratios decreased with increasing FG concentration (~6.7 at 5%, ~4.1 at 10%, and ~1.1 at 20% FG), indicating that the greater hydrated thickness of 20% gels mainly reflects their higher solid content rather than swelling.

- **New text (Line 623-635 – new method section): “Morphological characterization of robot-cast gels in dried states”.** The radius, thickness, and flatness of the fabricated hydrogels with cylindrical shapes were characterized in dried forms by optical profilometry. Hydrogel samples featuring 5%, 10%, and 20% w/v fish gelatin were cast on glass coverslips (VD12260Y1A.01, Knittel Glasbearbeitungs GmbH) using the scalable protocol previously described and left to crosslink overnight at 37 °C. The samples were then let dry for at least one day at room temperature (24 °C) and atmosphere. The samples in the dried state were then imaged with a Wyko NT9300 white light profilometer (Veeco) using a 20X 0.4 NA CF IC Epi Plan DI Mirau Interferometry Objective (Nikon) in vertical scanning interferometer (VSI) mode. The data was processed, analyzed, and 3D rendered using Vision 4.2 software (Veeco). At least three hydrogels per condition were imaged, and x- and y-cuts were taken in correspondence with the center of each gel to estimate thickness and flatness. Spherical aberration in the optical setup and substrate curvature were corrected using a second-order polynomial normalization using the thickness values of the glass substrate around the hydrogels.

Comment 3.5

(Page 14, Fig 4f) Nocodazole concentrations corresponding to the IC50 value can be shown on the curves if possible.

Response 3.5

Following the recommendation, we have modified Fig. 4f by adding the intercepts on both axes, which now clearly mark the IC50 value for nocodazole on the dose-response curve. To ensure consistency, we also applied a similar modification to the corresponding data for paclitaxel in the Supplementary Fig. S6f, where the intercepts are likewise indicated. We believe this addition improves the clarity of the figures and facilitates direct visualization of the IC50 values.

Modification 3.5

- **Old text (Line 255):** ... (f) Nocodazole dose-response curves on plastic (IC₅₀, 16.6 ng mL⁻¹) and hydrogel (IC₅₀, 18.1 ng mL⁻¹) substrates (n=12). Data were normalized concerning the initial confluency value (n=12). Image brightness and contrast were adjusted for printed visibility...
- **New text (Line 291-292):** ... (f) Nocodazole dose-response curves on plastic (IC₅₀, 16.6 ng mL⁻¹) and hydrogel (IC₅₀, 18.1 ng mL⁻¹) substrates (n=12). *To facilitate visualization of the half-maximal inhibitory concentration (IC₅₀), horizontal and vertical dotted lines mark the intercepts on the Y- and X-axes, respectively, corresponding to the IC₅₀ values for each condition.* Data were normalized concerning the initial confluency value (n=12). Image brightness and contrast were adjusted for printed visibility...
- **Old text SI (Line 77-79):** ... (f) Paclitaxel dose-response curves on plastic (IC₅₀, 1.6 ng mL⁻¹) and hydrogel (IC₅₀, 1.8 ng mL⁻¹) substrates (n=12). Data were normalized concerning the initial confluency value (n=12) ...
- **New text (Line 89-91):** ... (f) Paclitaxel dose-response curves on plastic (IC₅₀, 1.6 ng mL⁻¹) and hydrogel (IC₅₀, 1.8 ng mL⁻¹) substrates (n=12). *To help visualize the half-maximal inhibitory concentration (IC₅₀), horizontal and vertical dotted lines mark the intercepts on the Y- and X-axes, respectively, corresponding to the IC₅₀ values for each condition.* Data were normalized concerning the initial confluency value (n=12) ...

Comment 3.6

(Page 15 - line 275) *What would be the approximate % of cells of the total population that exhibited nuclear fragmentation and were excluded from analysis?*

Response 3.6

To address this question, we performed a manual count of cells displaying nuclear fragmentation across five representative fields of view (FOVs) for each treatment condition. For nocodazole (50 ng/mL), the percentage of cells with nuclear fragmentation ranged from 3% to 19%, with an overall mean of 11.2% ($\pm 5.9\%$) across the analyzed FOVs. For paclitaxel (12.5 ng/mL), the percentage ranged from 4% to 26%, with an overall mean of 12.2% ($\pm 8.7\%$).

Thus, on average, approximately 10–12% of the total cell population exhibited nuclear fragmentation and were therefore excluded from further analysis. These values are consistent across replicates and conditions, confirming that the excluded subpopulation represented a small but non-negligible fraction of the total.

Figure Supplementary S11. ... (e) Manual counts of cells exhibiting nuclear fragmentation were performed across five independent fields of view (FOVs) for each condition. Bars represent the mean percentage of cells with nuclear fragmentation (\pm SD) relative to the total cell population, while individual dots indicate values from each FOV. On average, ~11% of cells treated with nocodazole (50 ng/mL) and ~12% of cells treated with paclitaxel (12.5 ng/mL) displayed nuclear fragmentation and were excluded from subsequent analyses.

Modification 3.6

- **Old text (Line 278-280):** Furthermore, we observed nuclear fragmentation (Supplementary Fig. S10 - iv) and a comparable trend in the number of cells undergoing mitosis in paclitaxel-treated samples (Supplementary Fig. S10 - v).
- **New text (Line 315-318):** Furthermore, we observed nuclear fragmentation (Supplementary Fig. S10-iv), which accounted on average for ~10–12% of the total cell population across both treatments (see Supplementary Fig. S11), and a comparable trend in the number of cells undergoing mitosis in paclitaxel-treated samples (Supplementary Fig. S10-v).
- **Old text (Line 642-644):** The linking of daughter cells was performed manually. Manual corrections (~5%) were applied to recover daughter cells after mitosis, when FUCCI signals were briefly absent.

- **New text (Line 738-740):** ... were briefly absent... *Under high-concentration drug treatments, cells exhibiting nuclear fragmentation were identified by visual inspection across five representative fields of view, manually counted, and excluded from analysis.*

Comment 3.7

It is impressive that the authors have done several high-resolution imaging-based analyses on cells in both 96 and 384 well plates for several drug treatments and compared the results between cells cultured on plastic plates and hydrogels. However, some basic quantification on how much the single cell and nuclear morphologies differ in both cases may be useful to report as well.

Response 3.7

We performed a quantitative analysis of single-cell morphology by measuring the projected nuclear area under control conditions (0.1% DMSO) in both plastic and hydrogel substrates. The analysis was stratified by cell-cycle phases (EG1, G1, S/G2, and T) as identified in our imaging-based workflow.

As shown in the new figure Supplementary Fig. S11, we found that nuclear areas were overall comparable between plastic and hydrogel in EG1 and T phases (not significant), while significant differences emerged in G1 and S/G2, where cells on plastic substrates displayed larger spread areas compared to those on hydrogels ($p < 0.05$). These results highlight that substrate stiffness and composition can subtly, but measurably, influence single-cell morphology in a cell-cycle dependent manner.

Figure Supplementary S11. ... (f) Area of the nuclei was quantified under control conditions (0.1% DMSO, i.e. negative control) in cells cultured on plastic (grey) and hydrogel (blue) substrates and stratified by cell-cycle phase (EG1, G1, S/G2, T). Measurements were taken 48 h after seeding. Cells grown on plastic displayed significantly larger projected nuclear areas in G1 and S/G2 compared to hydrogel ($p < 0.05$), while no significant differences were observed in EG1 and T. Each dot represents the mean area value in each well; horizontal bars indicate mean \pm SEM.

We have now included these new findings in the revised manuscript, adding them at the end of paragraph describing fluorescence imaging in the 96-well plate experiment.

Modification 3.7

- **Old text (Line 280):** ... in paclitaxel-treated samples (Supplementary Fig. S10 - v).
- **New text (Line 318-325):** ... in paclitaxel-treated samples (Supplementary Fig. S10 - v). *In addition to drug-induced effects, we investigated whether substrate properties influenced single-cell morphology under control conditions (0.1% DMSO). By quantifying projected cell area 48 h after seeding, we observed that cells cultured on plastic exhibited significantly larger nuclear areas in G1 and S/G2 compared to those on hydrogels ($p < 0.05$), whereas no significant differences were detected in EG1 and T (Figure S11). These findings are consistent with the established principle that softer substrates restrict cell spreading relative to stiffer ones (Janmey et al., 2020), supporting the view that hydrogel mechanics can modulate cell morphology in a cell-cycle dependent manner.*
- **Old text (Line 642-644):** The linking of daughter cells was performed manually. Manual corrections (~5%) were applied to recover daughter cells after mitosis, when FUCCI signals were briefly absent. Under high-concentration drug treatments, cells exhibiting nuclear fragmentation were identified by visual inspection across five representative fields of view, manually counted, and excluded from analysis.
- **New text (Line 735-738):** ... were briefly absent. *Quantitative single-cell morphology was assessed by measuring the projected nuclear area under control conditions (0.1% DMSO) on both plastic and hydrogel substrates, using ROIs extracted from the segmentation; analyses were stratified by cell-cycle phase (EG1, G1, S/G2, T) as identified in the imaging workflow. Under high-concentration drug treatments ...*

New reference:

- P. A. Janmey et al., “Stiffness Sensing by Cells”, *Physiological Reviews*, 100:2, 695-724, (2020). DOI: <https://doi.org/10.1152/physrev.00013.2019>

Comment 3.8

(Page 26 - Line 464) *How did you specifically choose this mesh size? Was a mesh convergence analysis performed?*

Response 3.8

We thank the reviewer for raising this important point. To ensure that our results are mesh-independent, we performed a convergence analysis using the meniscus height at the wall as the reference parameter. The mesh refinements were applied using COMSOL's predefined mesh settings (“fine” to “extra fine”). As shown in the new figure below, the predicted meniscus height stabilizes with mesh density finer than $\sim 20 \text{ mm}^{-1}$, while the associated numerical error decreases below 5% and approaches zero for the finest mesh tested (42 mm^{-1}).

We have now included this analysis and clarification in the revised manuscript.

Figure caption: Top: Convergence analysis calculating meniscus height at the wall as a function of mesh size. Bottom: relative error compared to the finest tested mesh. Results converge for meshes finer than “fine” with an error < 5%.

Modification 3.8

- **Old text (Line 464-465):** The mesh was created using the” physics-controlled” option with an “extra fine” option for mesh size.
 - **New text (Line 533-536):** *The mesh was created using the “physics-controlled” option in COMSOL. A mesh convergence analysis based on the meniscus height at the wall confirmed that both “fine” and “extra fine” meshes yielded convergent solutions (<5% error), and therefore the “fine” option was selected for all reported simulations.*
-

We thank the editor and the reviewers for their thorough second-round evaluation of our work. Below, we provide detailed responses to all new comments and summarize the corresponding modifications made in this revised version of the manuscript.

Reviewer 2

Comment 2.1

Response 2.1 is not acceptable. And the references mentioned are too old (2 are from 2012). What was expected was comments/comparisons in regard to commercial bioprinters and their capabilities (e.g., Rastrum from Inventia Life Sciences). Then the rationale for choosing fish gelatin over more established materials such as PEG or collagen' was not addressed at all.

Response 2.1

We thank the reviewer for insisting on a more explicit comparison to commercial plate-based bioprinting and multiwell hydrogel systems. We misread the reviewer's meaning of "2.5D" in our first reply and discussed microfabricated substrates instead of droplet/plate bioprinting platforms. We have corrected that oversight. The Introduction now explicitly compares HYDRA also to commercial plate-based bioprinters (e.g., RASTRUM) and hydrogel solutions, and we added Supplementary Table S1 plus up-to-date citations. We also expanded and clarified the rationale for choosing cold-water fish gelatin (FG) crosslinked with microbial transglutaminase (TG). Key points added, summarized below:

Scope and niche. HYDRA is optimized for planar, imaging-first applications (10–50 μm continuous films in standard 96/384 wells) compatible with high-NA objective access and automated liquid-handling workflows. This contrasts with droplet-on-demand 3D plate printers (e.g., RASTRUM / Inventia), which deposit cell-laden volumetric droplets optimized for volumetric assays and spheroid/organoid workflows but that produce mm-scale structures with reduced single-plane optical access and different perfusion/assay constraints. We make explicit that these are complementary, not redundant, approaches.

Throughput and automation practicalities. As summarized in Table S1, HYDRA operates with standard air-displacement pipetting and room-temperature handling, without the need for cooled reservoirs or custom printheads. By contrast, many 3D plate printers and hydrogel systems (e.g., RASTRUM, TrueGel3D HTS, PEG-based or HA-based biomaterials) require temperature-controlled dispensing or custom nozzles, limiting scalability for routine high-throughput workflows. The new table highlights gel thickness, optical accessibility, and imaging modality across platforms.

Optical and assay tradeoffs. We clarified that thick photocured PEG plates and mm-scale 3D gels provide useful volumetric biology (invasion, 3D viability), but their thickness and refractive index mismatches limit high-NA imaging and increase meniscus artefacts. Conversely, ultrathin LbL/PEM films are optically flat but are too thin to deliver cell-relevant soft mechanics at the cell–substrate length scales HYDRA targets. HYDRA intentionally occupies the intermediate imaging-first soft-substrate niche.

Rationale for material choice. We expanded the Introduction to justify our selection:

- Cold-water fish gelatin (FG) remains liquid at room temperature, permitting accurate air-displacement dispensing and re-aspiration without chilled printheads or thermal control, which simplifies integration with standard HTS robotic platforms.

- Microbial transglutaminase (TG) enables enzymatic crosslinking to physiological stiffness ranges without UV exposure or free-radical initiators; this reduces phototoxicity and autofluorescence risk compared with photocured systems and yields thin, mechanically relevant coatings compatible with high-NA fluorescence imaging.
- The FG + TG combination produces flat, low-meniscus coatings when dispensed as thin films, improving imaging uniformity and compatibility with oil-immersion objectives and automated autofocus.

References and currency. We replaced/updated older references and added recent literature and product notes to support these comparisons; all entries in Table S1 include full citation information.

Modification 2.1

- [1] **Old text:** However, as with Brooks et al., the resulting gels are several hundred micrometres thick, which restricts high-resolution imaging to planes near the well bottom and makes them unsuitable for 2D mechanobiology assays that require a flat, optically accessible surface²⁸...
- [2] **New Text (Line 99-105):** ...that require a flat, optically accessible surface²⁸. *Commercial 2.5D and 3D systems highlight the same trade-off: pre-cast PEG matrices (e.g., TrueGel 3D-HTS, 0.5–1 mm) and bioprinted constructs such as RASTRUM (Inventia Life Sciences) enable high-throughput encapsulation of cell-laden gels but remain hundreds of micrometres thick and optimized for volumetric or biomarker assays rather than single-plane imaging^{28,32,33}. Together these approaches demonstrate the lack of a platform that combines optical flatness and biomimicry in a high-throughput format, while maintaining compatibility with standard plate geometry and imaging (Supplementary Table S1).*

New reference:

33. Utama, R. H. et al. A 3D Bioprinter Specifically Designed for the High-Throughput Production of Matrix-Embedded Multicellular Spheroids. *iScience* 23, (2020).

- [3] **Old text:** HYDRA works by robotically dispensing a sub-contact volume of hydrogel precursor solution, carefully preventing it from contacting well sidewalls. Immediate controlled re-aspiration pins the solution's contact line through contact angle hysteresis, yielding a thin, uniform precursor layer...
- [4] **New text (Line 111-120):** ...yielding a thin, uniform precursor layer. *Among available hydrogel chemistries, we selected cold-water fish gelatin (FG) for its favorable handling and biological compatibility. FG remains fluid at room temperature, allowing precise robotic dispensing and re-aspiration, while microbial transglutaminase (TG) enables mild enzymatic crosslinking without UV or heating^{34,35}. Unlike synthetic polymers such as PEG that require surface functionalization, gelatin is inherently cell-adhesive, supporting rapid attachment and spreading^{36,37}. Gelatin is also substantially order of magnitude less expensive than common GelMA or collagen (~€0.5 g⁻¹ vs €100s g⁻¹ for GelMA/collagen, current catalog prices from suppliers), an important factor for large-scale screening. These rheological, biochemical, and economic advantages make FG well suited to form planar, micrometric hydrogel films reproducibly within standard multiwell plates.*

New references:

35. Yang, G. et al. Enzymatically crosslinked gelatin hydrogel promotes the proliferation of adipose tissue-derived stromal cells. *PeerJ* 4, e2497 (2016).

36. Singh, S. P., Schwartz, M. P., Lee, J. Y., Fairbanks, B. D. & Anseth, K. S. A peptide functionalized poly(ethylene glycol) (PEG) hydrogel for investigating the influence of biochemical and biophysical matrix properties on tumor cell migration. *Biomater. Sci.* 2, 1024–1034 (2014).

37. Lu, P. et al. Harnessing the potential of hydrogels for advanced therapeutic applications: current achievements and future directions. *Signal Transduction and Targeted Therapy* 9, 166 (2024).

Supplementary Table S1. Hydrogel-based plate formats: throughput, imaging compatibility, strengths, and limitations.

Platform	Typical Throughput	Hydrogel Thickness / Geometry	Imaging Compatibility	Primary Use / Strength	Key Limitations
HYDRA (this work)	96–384 wells	10–50 μm flat film (meniscus-free)	High-content imaging compatible	High-throughput 2D soft-substrate assays; imaging-optimized	No encapsulation (2D only)
RASTRUM (Inventia)¹	96–384 wells	200–1000 μm printed 3D microtissues	Confocal/3D imaging	3D cell-laden bioprinting for drug response	Thick constructs; low optical clarity for 2D imaging
TrueGel3D HTS (Merck)²	96 wells	500–1000 μm bulk PEG gels	Confocal/3D imaging	3D invasion/viability assays	Meniscus curvature; poor single-plane imaging
PEG-based biomaterials (Brooks 2018)³	96 wells	500–1000 μm	Confocal/3D imaging	Live/Dead Cell viability assays	Surface imaging limited by thickness
LbL polyelectrolyte films (Machillot 2018)⁴	96 wells	10 nm–1 μm	Excellent optical flatness	Surface chemistry tuning; mechanobiology	Too thin to mask stiff substrate
LM-Well insert (Milton 2025)⁵	96 wells (multi-niche)	Multiple small flat gels	Imaging-friendly regions	Co-culture or organoid interface studies	Manual insert handling; moderate throughput
2D/3D Hydrogel platforms (Skelton 2024)⁶	96 wells	100–300 μm molded gels	Compatible with plate readers & high-content imaging	Multiplexed viscoelastic matrices	Assembled plate; moderate automation
LigHTS (Enrico 2025)⁷	384–1536 wells	10–60 μm photopatterned GelMA	High-content imaging compatible	Ultra-HTS patterned hydrogel coatings	Specialized photochemistry equipment

Supplementary table references:

- [1] **A 3D Bioprinter Specifically Designed for the High-Throughput Production of Matrix-Embedded Multicellular Spheroids.** Robert H. Utama, Lakmali Atapattu, Aidan P. O'Mahony, Christopher M. Fife, Jongho Baek, Théophile Allard, Kieran J. O'Mahony, Julio C.C. Ribeiro, Katharina Gaus 4 5, Maria Kavallaris, J. Justin Gooding. (2020). iScience. Doi: <https://doi.org/10.1016/j.isci.2020.101621>
- [2] **Soft Hydrogels Featuring In-Depth Surface Density Gradients for the Simple Establishment of 3D Tissue Models for Screening Applications.** Ning Zhang, Vincent Milleret, Greta Thompson-Steckel, Ning-Ping Huang, J. Vörös, Benjamin R. Simona, Martin Ehrbar. (2017) SLAS Discovery. Doi: <https://doi.org/10.1177/2472555217693191>.
- [3] **Complementary, Semiautomated Methods for Creating Multidimensional PEG-Based Biomaterials.** Brooks EA, Jansen LE, Gencoglu MF, Yurkevicz AM, Peyton SR. (2018). ACS Biomater Sci Eng. Doi: <https://doi.org/10.1021/acsbiomaterials.7b00737>.
- [4] **Automated Buildup of Biomimetic Films in Cell Culture Microplates for High-Throughput Screening of Cellular Behaviors.** Machillot P, Quintal C, Dalonneau F, Hermant L, Monnot P, Matthews K, Fitzpatrick V, Liu J, Pignot-Paintrand I, Picart C. (2018). Adv Mater. Doi: <https://doi.org/10.1002/adma.201801097>.
- [5] **Building multiple microenvironmental niches using a customizable 3D printed well insert.** Milton, Laura & Kasetsirikul, Surasak & Catano, Jorge & Hilmi, F & Zhou, Zeheng & Molley, Thomas & Kilian, Kristopher & Ong, Louis & Chirnside, James & Byrom, Nicholas & Balshaw, Georgia & Liang, Sammy & Bray, Laura & Hutmacher, Dietmar & Meinert, Christoph & Toh, Yi-Chin. (2025). Lab on a chip. Doi: <https://doi.org/10.1039/D5LC00753D>.
- [6] **Modular Multiwell Viscoelastic Hydrogel Platform for Two- and Three-Dimensional Cell Culture Applications.** Mackenzie L. Skelton, James L. Gentry, Leilani R. Astrab, Joshua A. Goedert, E. Brynn Earl, Emily L. Pham, Tanvi Bhat, and Steven R. Caliari. (2024). ACS Biomaterials Science & Engineering. Doi: <https://pubs.acs.org/doi/10.1021/acsbiomaterials.4c00312>.
- [7] **LIGHTS: Massively Parallel Biomimetic Photo-Functionalization for Imaging-Based Ultra-High-Throughput Screening.** Alessandro Enrico, Sara Rigolli, Julius Zimmermann, Melissa Pezzotti, Eloisa Torchia, Moises Di Sante, Ferdinando Auricchio, Francesco S. Pasqualini. (2025). BioRxiv. Doi: <https://doi.org/10.1101/2025.10.23.683892>.

Comment 2.2

Response 2.2 did not answer the problem.

Response 2.2

From the start, the Results section aimed to demonstrate feasibility: that HYDRA can run high-content/throughput, imaging-based pharmacology experiments on thin, mechanically defined hydrogel films. We acknowledge that the reviewer perceived an implicit claim that HYDRA produces wholesale potency re-ranking; that was a misunderstanding we worked to fix. Following the first revision, we made this scope explicit and added literature context showing that mechano-sensitivity is cell-line dependent but that our results match what other people have reported for the same (and similar cell types), only collected in high-throughput. This is precisely the claim our data is meant to support. I am not sure there is another problem to answer to.

Our goal in Result sections 2.3–2.4 is to demonstrate feasibility of imaging-based HTS on thin, soft hydrogels, not to claim wholesale potency re-ranking. We now say this explicitly and have tightened the claim: HYDRA preserves pharmacological IC₅₀s (e.g., nocodazole 16.6 ng mL⁻¹ on plastic vs 18.1 ng mL⁻¹ on FG; paclitaxel 1.6 vs 1.8 ng mL⁻¹) while increasing proliferative headroom and imaging quality (e.g., ~2× higher final confluency on FG at matched dose; Fig. 4d–f; S6–S7). The small IC₅₀ deltas we observe are concordant with the literature for 2D stiffness perturbations in HaCaT/HeLa systems (e.g., HeLa paclitaxel 2.3 nM at 1 kPa vs 1.9 nM on glass; MCF-7 larger shifts; PC3 minimal), underscoring that mechano-sensitivity is cell-

line dependent rather than a failure of the thin-film approach. We therefore refrain from over-interpreting potency changes and position HYDRA as the pragmatic, imaging-optimized step-change toward more physiological early HTS, consistent with the revised Discussion and recent references.

Modification 2.2

- **Old text:** The adhesion and proliferation of engineered HaCaT cells that we observed on HYDRA gels are consistent with the well-established behavior of this cell line, which is known to maintain strong adhesion, proliferative capacity, and compatibility with imaging across a range of substrates⁵³. In this proof-of-concept study with HaCaT keratinocytes, chosen for their stable cell-cycle reporters, the IC₅₀ differences between plastic and hydrogel substrates were modest (1.6 vs. 1.8 nM). Similar small shifts have been reported in HeLa cells (2.3 nM at 1 kPa vs. 1.9 nM on glass⁷³, while other lines show larger or negligible effects depending on stiffness⁷⁴. These findings indicate that HYDRA preserves drug responsiveness while offering a physiologically relevant and reproducible culture substrate.
- **New text (Line 454-464):** The adhesion and proliferation of engineered HaCaT cells that we observed on HYDRA gels are consistent with the well-established behavior of this cell line, which is known to maintain strong adhesion, proliferative capacity, and compatibility with imaging across a range of substrates⁵³. *In this proof-of-concept study, the goal was to demonstrate feasibility rather than functional re-ranking of drug potency. We intentionally selected HaCaT keratinocytes for their stable cell-cycle reporters to establish compatibility of imaging-based pharmacological assays on thin, mechanically defined hydrogels. The IC₅₀ differences between plastic and hydrogel substrates were modest (1.6 vs. 1.8 nM), but consistent with prior reports in similar epithelial systems (e.g., HeLa cells: 2.3 nM at 1 kPa vs. 1.9 nM on glass⁷³), while other lines show larger or negligible effects depending on stiffness⁷⁴. These results demonstrate that HYDRA preserves canonical drug responsiveness while providing a reproducible, biomimetic substrate for high-content pharmacological imaging. Future work will extend this framework to stiffness-sensitive models such as hiPSC-derived cardiomyocytes (Ribeiro et al., 2015, Pasqualini et al., 2018), to quantitatively assess whether the biomimetic mechanical environment provided by HYDRA enhances predictive power in large-scale pharmacological screens.*

New references:

- A.J.S. Ribeiro, et al., Contractility of single cardiomyocytes differentiated from pluripotent stem cells depends on physiological shape and substrate stiffness, Proc. Natl. Acad. Sci. U.S.A. <https://doi.org/10.1073/pnas.1508073112> (2015).
- Pasqualini FS, et al., Traction force microscopy of engineered cardiac tissues. PLoS One. <https://doi.org/10.1371/journal.pone.0194706> (2018).

Comment 2.3

The additional figure demonstrates how heterogeneous the system and how non-reproducible it is. Then the claim that ‘, iii) a comparison with existing platforms across throughput levels has been added’ was meant as an actual comparison, although this is acknowledged that this could be difficult. As least a literature comparison should be provided.

Response 2.3

The additional z-stack the reviewer cites was a deliberate “meniscus control” produced without HYDRA to illustrate why meniscus curvature undermines single-plane imaging and yields edge-biased cell density; the revised legend now states “Hydrogel casting (non-HYDRA) in high-throughput plates” to avoid confusion. In contrast, our automated QC over 384-well plates shows ~91 % planar hydrogels, with only 6 % wall-wet and 3 % concave wells (Fig. 6a), and high-NA live/confocal imaging across dozens of FOVs in 384-well plates (Fig. 6b–d). To meet the request for an “actual comparison,” we have added Supplementary Table S1 (see Response 2.1) that collates peer-reviewed and commercial 2.5D/3D plate formats (such as RASTRUM, TrueGel3D-HTS, thick PEG in wells, LbL PEMs) against HYDRA on (i) throughput, (ii) gel thickness/flatness, (iii) imaging/readouts, and (iv) typical use-case. This table makes explicit that thick 3D plates (hundreds of μm) and nm– μm LbL films serve different objectives but do not provide the flat, 10–50 μm , full-well coatings needed for single-plane, high-NA, 96/384-well imaging; conversely, HYDRA does, as reflected in the QC and imaging data now foregrounded in 2.5 and Fig. 6.

Modification 2.3

Supplementary Figure S4: *...(d-e) Hydrogels obtained by conventional liquid casting, displaying high curvature at well edges in contrast with meniscus-free hydrogels produced using HYDRA. (d) Representative z-stack reconstruction of a hydrogel with full meniscus inside a high-throughput well, displayed in orthogonal views (xy, xz, yz). The hydrogel containing fluorescent beads is shown in magenta, while cells (actin) appear in gray. (e) Zoom-in of the xy plane from panel d, highlighting the curved meniscus across the entire well area.*